# Multiple preferred escape trajectories are explained by a geometric model incorporating prey's turn and predator attack endpoint

Yuuki Kawabata[1]*, Hideyuki Akada[2], Ken-ichiro Shimatani[3], Gregory Naoki Nishihara[4], Hibiki Kimura[1], Nozomi Nishiumi[1,5], Paolo Domenici[6,7]

[1]Graduate School of Fisheries and Environmental Sciences, Nagasaki University, Nagasaki, Japan; [2]Faculty of Fisheries, Nagasaki University, Nagasaki, Japan; [3]The Institute of Statistical Mathematics, Tachikawa, Japan; [4]Institute for East China Sea Research, Organization for Marine Science Technology, Nagasaki University, Nagasaki, Japan; [5]National Institute for Basic Biology, Okazaki, Japan; [6]CNR-IAS, Località Sa Mardini, Oristano, Italy; [7]CNR-IBF, Area di Ricerca San Cataldo, Pisa, Italy

*For correspondence:
yuuki-k@nagasaki-u.ac.jp

Competing interest: The authors declare that no competing interests exist.

**Abstract** The escape trajectory (ET) of prey – measured as the angle relative to the predator's approach path – plays a major role in avoiding predation. Previous geometric models predict a single ET; however, many species show highly variable ETs with multiple preferred directions. Although such a high ET variability may confer unpredictability to avoid predation, the reasons why animals prefer specific multiple ETs remain unclear. Here, we constructed a novel geometric model that incorporates the time required for prey to turn and the predator's position at the end of its attack. The optimal ET was determined by maximizing the time difference of arrival at the edge of the safety zone between the prey and predator. By fitting the model to the experimental data of fish *Pagrus major*, we show that the model can clearly explain the observed multiple preferred ETs. By changing the parameters of the same model within a realistic range, we were able to produce various patterns of ETs empirically observed in other species (e.g., insects and frogs): a single preferred ET and multiple preferred ETs at small (20–50°) and large (150–180°) angles from the predator. Our results open new avenues of investigation for understanding how animals choose their ETs from behavioral and neurosensory perspectives.

## Editor's evaluation

This article will be of interest to researchers working on predator-prey interactions in the fields of biomechanics and neurosensory biology. It presents a valuable mathematical model that outputs possible escape trajectories given parameters relevant to the predator-prey system of interest. The premise of the modeling is attractive, as it includes the time required for prey to turn.

## Introduction

When exposed to sudden threatening stimuli such as ambush predators, most prey species initiate escape responses that include turning swiftly and accelerating away from the threat. The escape responses of many invertebrate and lower vertebrate species are controlled by giant neurons that ensure a short response time (*Bullock, 1984*). Many previous studies have focused on two behavioral traits that are fundamental for avoiding predation: when to escape (i.e., flight initiation distance,

**eLife digest** When a prey spots a predator about to pounce, it turns swiftly and accelerates away to avoid being captured. The initial direction the prey chooses to take – known as its escape trajectory – can greatly impact their chance of survival.

Previous models were able to predict the optimal direction an animal should take to maximize its chances of evading the predator. However, experimental data suggest that prey actually tend to escape via multiple specific directions, although why animals use this approach has not been clarified. To investigate this puzzle, Kawabata et al. built a new mathematical model that better represents how prey and predators interact with one another in the real world.

Unlike past models, Kawabata et al. incorporated the time required for prey to change direction and only allowed the predators to move toward the prey for a limited distance. By including these two factors, they were able to reproduce the escape trajectories of real animals, including a species of fish, as well as species from other taxa such as frogs and insects.

The new model suggests that prey escape along one of two directions: either by moving directly away from the predator in order to outrun its attack, or by dodging sideways to avoid being captured. Which strategy the prey chooses has some elements of unpredictability, which makes it more difficult for predators to adjust their capturing method.

These findings shed light on why escaping in multiple specific directions makes prey harder to catch. The model could also be extended to test the escape trajectories of a wider variety of predator and prey species, which may avoid capture via different routes. This could help researchers better understand how predators and prey interact with one another. The findings could also reveal how sensory information (such as sound and sight) associated with the threat of an approaching predator is processed and stimulates the muscle activity required to escape in multiple different directions.

which is measured as the distance from the predator at the onset of escape) and where to escape (i.e., escape trajectory [ET], which is measured as the angle of escape direction relative to the stimulus direction) (*Cooper, Jr and Blumstein, 2015*). Previous studies have investigated the behavioral and environmental contexts affecting these variables (*Meager et al., 2006*; *Arnott et al., 1999*; *Bateman and Fleming, 2014*; *Hein et al., 2018*; *Broom and Ruxton, 2005*; *Cooper et al., 2003*), because they largely determine the success or failure of predator evasion (*Walker et al., 2005*; *Shifferman and Eilam, 2004*; *Camhi et al., 1978*; *Kimura and Kawabata, 2018*; *Dangles et al., 2006*), and hence the fitness of the prey species. A large number of models on how animals determine their flight initiation distances have been formulated and tested by experiments (*Cooper, Jr and Blumstein, 2015*). Although a number of models have also been developed to predict animal ETs (*Arnott et al., 1999*; *Weihs and Webb, 1984*; *Domenici, 2002*), there are still some unanswered questions about how the variability of the observed ETs is generated.

Two different escape tactics (and their combination) have been proposed to enhance the success of predator evasion (*Jensen, 2018*; *Domenici et al., 2011a*): the optimal tactic (deterministic), which maximizes the distance between the prey and the predator (*Figure 1A*; *Arnott et al., 1999*; *Weihs and Webb, 1984*; *Domenici, 2002*; *Soto et al., 2015*), and the protean tactic (stochastic), which maximizes unpredictability to prevent predators from adjusting their strike trajectories accordingly (*Figure 1B*; *Humphries and Driver, 1970*; *Jones et al., 2011*; *Richardson et al., 2018*; *Moore et al., 2017*). Previous geometric models, which formulate optimal tactics, predict a single ET that depends on the relative speeds of the predator and the prey (*Arnott et al., 1999*; *Weihs and Webb, 1984*; *Domenici, 2002*; *Soto et al., 2015*), and additionally, predator's turning radii and sensory-motor delay in situations where the predator can adjust its strike path (*Howland, 1974*; *Corcoran and Conner, 2016*; *Martin et al., 2022*). The combination of the optimal tactic (formulated by previous geometric models), which predicts a specific single ET, and the protean tactic, which predicts variability, can explain the ET variability within a limited angular sector that includes the optimal ET (*Figure 1C*). However, the combination of the two tactics cannot explain the complex ET distributions reported in empirical studies on various taxa of invertebrates and lower vertebrates (reviewed in *Domenici et al., 2011b*). Whereas some animals exhibit unimodal ET patterns that satisfy the prediction of the combined tactics or optimal tactic with behavioral imprecision (e.g., *Cooper, 2006*), many animal

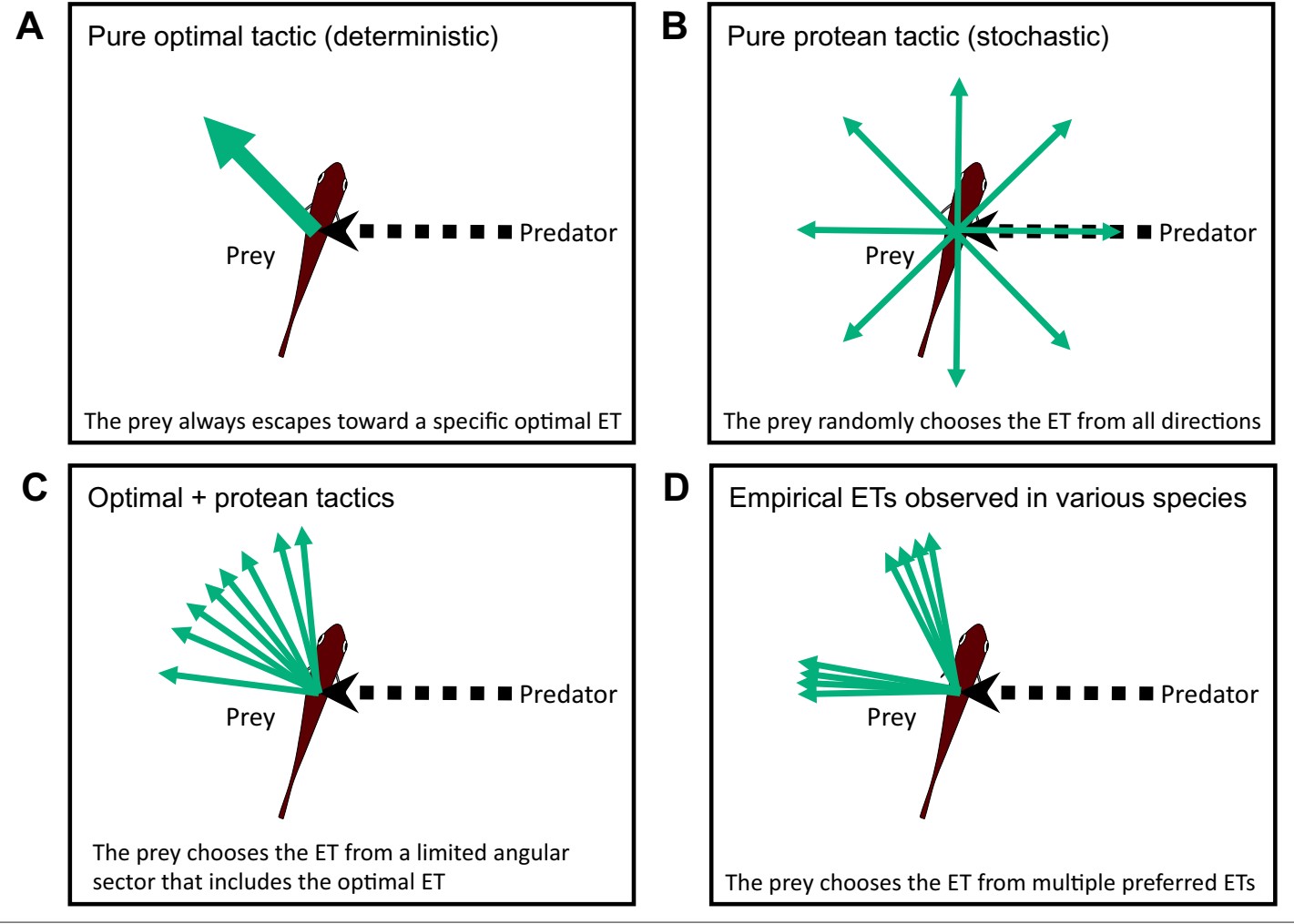

**Figure 1.** Conceptual diagram showing the different tactics for escape trajectories (ETs). (**A**) The pure optimal tactic, which predicts a specific optimal ET. (**B**) The pure protean tactic, which predicts a random ET from all directions. (**C**) The combination of optimal and protean tactics, which predicts an ET selected randomly (or with a specific probability distribution) from a limited angular sector that includes the optimal ET. (**D**) The multiple preferred ETs, empirically observed in various species. Please also see *Domenici et al., 2011a*, for the review on potential ETs.

species show multimodal ETs within a limited angular sector (esp., 90–180°) (*Figure 1D*) (e.g., *Arnott et al., 1999*; *Bateman and Fleming, 2014*; *Domenici and Blake, 1993*). To explore the discrepancy between the predictions of the models and empirical data, some researchers have hypothesized mechanical/sensory constraints (*Domenici et al., 2011a*; *Domenici et al., 2008*); however, the reasons why certain animal species prefer specific multiple ETs remain unclear.

Multiple preferred ETs of prey can result from situations in which animals choose one behavior from multiple options. Previous work carried out in the field of human and animal psychology on the choice of a particular behavioral strategy out of a number of options has proposed a principle called 'matching law'. According to this principle, the probability of a certain behavior to occur is related to the proportion of rewards obtained (*Reed and Kaplan, 2011*; *Poling et al., 2011*; *McDowell, 2013*; *Houston et al., 2021*). This is in contrast to a purely optimal tactic, where animals should always choose the best option (i.e., the highest rewards obtained) (*Houston et al., 2021*; *Fawcett et al., 2013*). Arguably, the field of predator-prey interactions has the potential to benefit from an analytical interpretation based on the matching law, because the multiple ETs available to the prey set a scenario similar to the multiple behavioral options considered in previous work analyzed using this principle. In line with this approach, the probability with which a prey chooses a particular ET can be related to the rewards (chances of survival) of each ET option calculated from a predator-prey geometric model.

In previous geometric models, the prey was assumed to instantaneously escape in any direction, irrespective of the prey's initial body orientation relative to the predator's approach path (hereafter, initial orientation) (*Arnott et al., 1999*; *Weihs and Webb, 1984*; *Domenici, 2002*). However, additional time is required for changing the heading direction (i.e., turn); therefore, a realistic model needs to take into account that the predator can approach the prey while the prey is turning (*Kimura and Kawabata, 2018*). Additionally, in previous models, attacking predators were assumed to move for an infinite distance at a constant speed (*Arnott et al., 1999*; *Weihs and Webb, 1984*; *Domenici, 2002*). However, the attacks of many real predators, especially ambush ones, end at a certain distance from initial positions of the prey (*Webb and Skadsen, 1980*; *Fouts and Nelson, 1999*; *Anderson, 1993*). Therefore, we constructed a geometric model that incorporates two additional factors: the time required for the prey to turn and the endpoint of the predator attack. First, using a fish species as a model, we tested whether our model could predict empirically observed multimodal ETs. Second, by calculating the chances of survival of each ET option from our model, we investigated how the prey fish chose a given ET from multiple options. Third, by extending the model, we tested whether other patterns of empirical ETs could be predicted: unimodal ETs and multimodal ETs directed at small (20–50°) and large (150–180°) angles from the predator's approach direction. The biological implications resulting from the model and experimental data are then discussed within the frameworks of predator-prey interactions and behavioral decision-making.

## Model

We revised the previous model proposed by *Domenici, 2002*; *Paglianti and Domenici, 2006* (*Figure 2A*) and the model proposed by *Corcoran and Conner, 2016* (*Appendix 1—figure 1A*). Other previous models (*Arnott et al., 1999*; *Weihs and Webb, 1984*; *Soto et al., 2015*; *Martin et al., 2022*) made predictions similar to those of Domenici's model or those of Corcoran's model, although they used different theoretical approaches. In Domenici's model, the predator with a certain width (i.e., the width of a killer whale's tail used as a weapon to catch prey) directly approaches the prey, and the prey (the whole body) should enter the safety zone before the predator reaches that entry point. In this model, the prey can instantaneously escape in any direction, and the predation threat moves linearly and infinitely. Corcoran's model is based on the same principle as Domenici's model, but includes the concept that the predator (i.e., a bat) can adjust the approach path up to its minimum turning radius. Thus, Domenici's model can be regarded as a special case of Corcoran's model when the turning radius of the predator is infinitely large. These models are based on the escape response of the horizontal plane, which is realistic for many fish species as well as terrestrial and benthic species that move on substrates. They can also be applied to aerial animals such as moths escaping from bats because many predator-prey interactions are approximately two-dimensional in a local spatial scale (*Corcoran and Conner, 2016*; *Fabian et al., 2018*). Hereafter, we explain the modification of Domenici's model (a special case of Corcoran's model) because the data on previously published predator-prey experiments on the same species of prey and predator in our experiment (*Kimura and Kawabata, 2018*) show that the predator does not adjust the strike path during the attack (*Figure 2—figure supplement 1*, adjusted angle = 1.0 ± 6.6° [mean ± s.d.], *n*=5), and thus the number of parameters to estimate can be reduced. See Appendix 1 for details of the modified version of Corcoran's model.

In our new model (*Figure 2B*), two factors are added to the previous Domenici's model: the time required for the prey to turn and the endpoint of the predator attack. We assume that a prey with a certain initial orientation $\beta$ (spanning 0–180°, where 0° and 180° correspond to being attacked from front and behind, respectively) evades a sudden predation threat. Most prey species respond to the attack by turning at an angle $\alpha$, and the ET results from the angular sum of $\alpha$ and $\beta$. ETs from the left and right sides were pooled and treated as though they were stimulated from the right side (*Figure 2—figure supplement 2*; see 'Definition of the angles' in Materials and methods for details).

When the prey's CoM at the onset of its escape is located at point (0, 0), the trajectory of the CoM ($X_{prey}$, $Y_{prey}$) is given by:

$$Y_{prey} = X_{prey}\tan\left(\alpha + \beta\right) \tag{1}$$

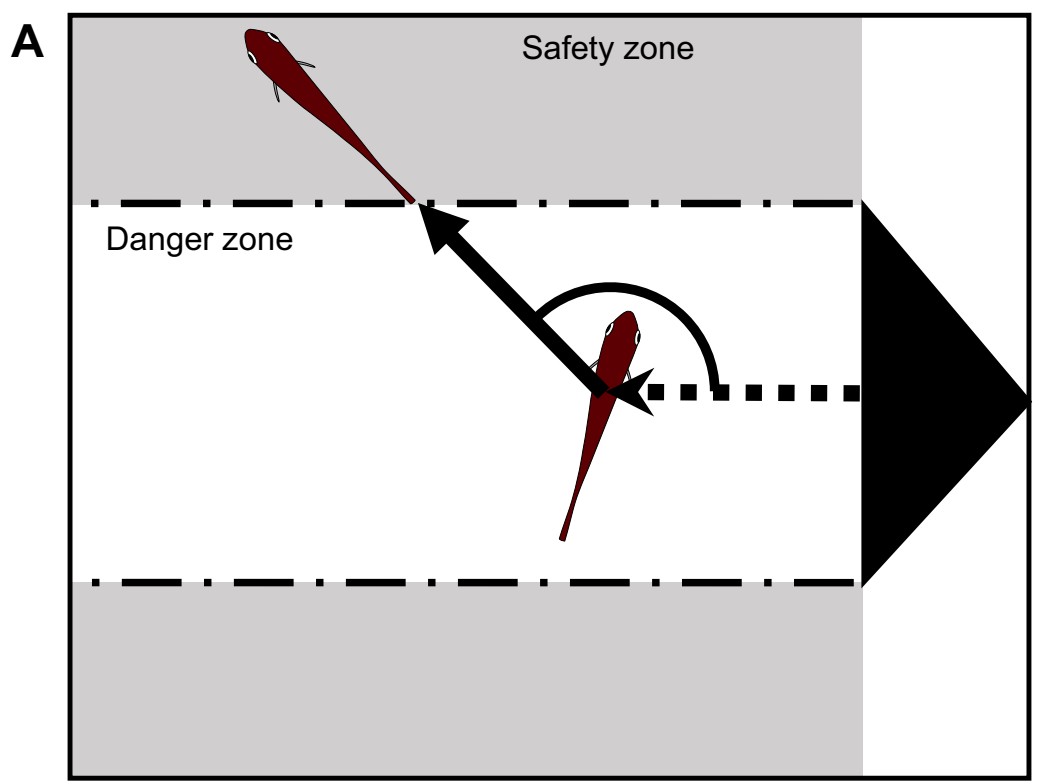

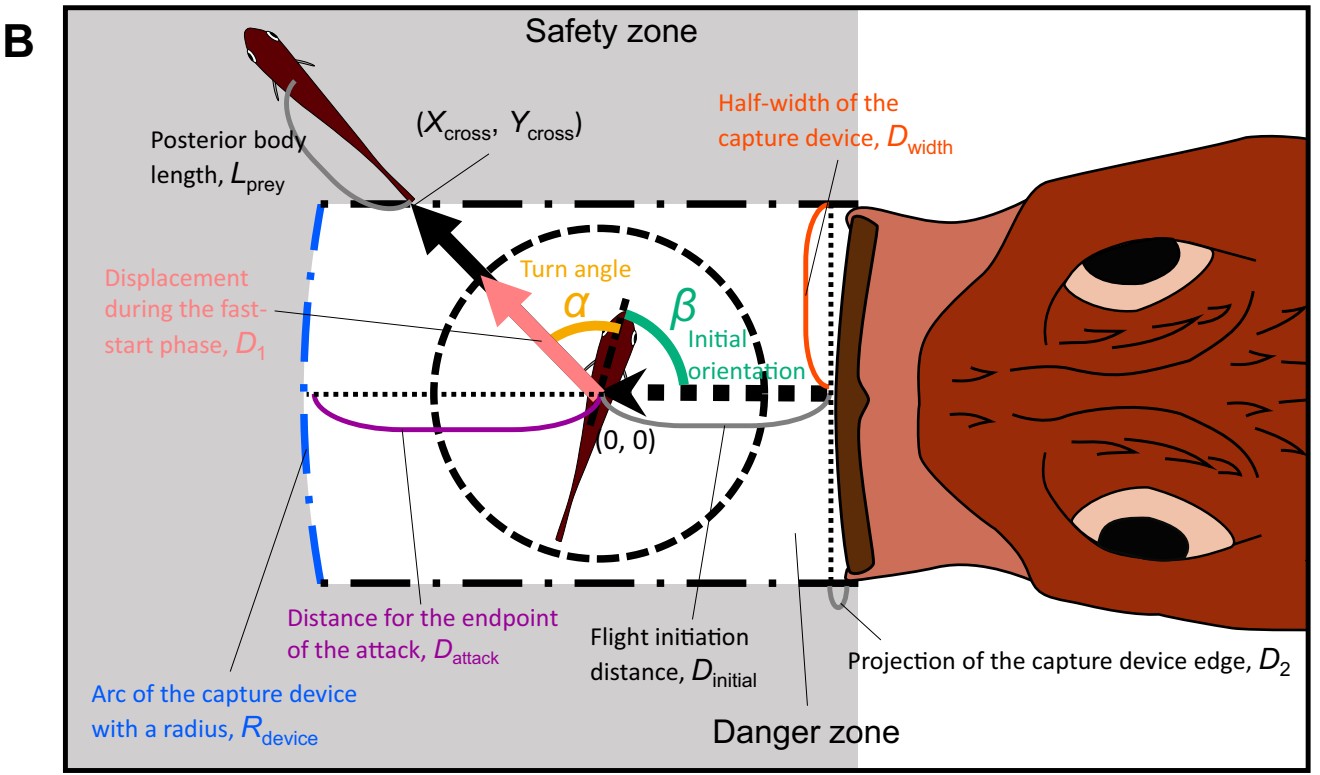

**Figure 2.** Proposed geometric models for animal escape trajectories. (**A**) A previous geometric model proposed by *Domenici, 2002*; *Paglianti and Domenici, 2006*. The predation threat with a certain width (the tail of a killer whale, represented by the black triangle) directly approaches the prey, and the prey should reach the safety zone (a grey area) outside the danger zone (white area) before the threat reaches that point. In this model, the prey can instantaneously escape in any direction, and the predation threat moves linearly and infinitely. (**B**) Two factors are added to Domenici's model:

*Figure 2 continued on next page*

*Figure 2 continued*
the endpoint of the predator attack, and the time required for the prey to turn. ($X_{\text{cross}}$, $Y_{\text{cross}}$) denotes the x and y coordinates of the crossing point of the escape path and the safety zone edge.

The online version of this article includes the following figure supplement(s) for figure 2:

**Figure supplement 1.** Schematic drawing of how the adjusted angle of the predator ($\theta$) was measured.

**Figure supplement 2.** Schematic drawing of angular variables.

The edge of the safety zone is determined by the half-width of the predator capture device (e.g., mouth) $D_{\text{width}}$, the distance between the prey's initial position and the tip of the predator capture device at the end of the predator attack $D_{\text{attack}}$, and the shape of the predator's capture device at the moment of attack, which is approximated as an arc with a certain radius $R_{\text{device}}$. The projection of the predator's capture device edge along the edge of the sideways safety zone $D_2$ can be expressed as:

$$D_2 = R_{\text{device}} \left\{ 1 - \cos\left(\sin^{-1}\frac{D_{\text{width}}}{R_{\text{device}}}\right) \right\} \tag{2}$$

The ET toward the upper-left corner of the danger zone $\theta_{\text{corner}}$ can be expressed as:

$$\theta_{\text{corner}} = \tan^{-1}\frac{D_{\text{width}}}{D_2 - D_{\text{attack}}} \tag{3}$$

The x and y coordinates of the safety zone edge ($X_{\text{safe}}$, $Y_{\text{safe}}$) are given by:

$$\begin{cases} Y_{\text{safe}} = D_{\text{width}}, \quad \alpha + \beta < \theta_{\text{corner}} \\ \left(X_{\text{safe}} + D_{\text{attack}} - R_{\text{device}}\right)^2 + Y_{\text{safe}}^2 = R_{\text{device}}^2, \quad \alpha + \beta \geq \theta_{\text{corner}} \end{cases} \tag{4}$$

From *Equation 1* to *Equation 4*, the x and y coordinates of the crossing point of the escape path and the safety zone edge ($X_{\text{cross}}$, $Y_{\text{cross}}$) are given by a function of $D_{\text{width}}$, $D_{\text{attack}}$, $R_{\text{device}}$, and $\alpha+\beta$.

The prey can escape from the predator when the time required for the prey to enter the safety zone ($T_{\text{prey}}$) is shorter than the time required for the predator's capture device to reach that entry point ($T_{\text{pred}}$). Therefore, the prey is assumed to maximize the difference between the $T_{\text{pred}}$ and $T_{\text{prey}}$ ($T_{\text{diff}}$). To incorporate the time required for the prey to turn, $T_{\text{prey}}$ was divided into two phases: the fast-start phase, which includes the time for turning and acceleration ($T_1$), and the constant speed phase ($T_2$). This assumption is consistent with the previous studies (*Domenici and Blake, 1991*; *Danos and Lauder, 2012*; *Fleuren et al., 2018*) and was supported by our experiment (see *Figure 4—figure supplement 1*). Therefore:

$$T_{\text{prey}} = T_1 + T_2 \tag{5}$$

For simplicity, the fish was assumed to end the fast-start phase at a certain displacement from the initial position in any $\alpha$ ($D_1$; the radius of the dotted circle in *Figure 2B*) and to move at a constant speed $U_{\text{prey}}$ to cover the rest of the distance (toward the edge of the safety zone $\sqrt{X_{\text{cross}}^2 + Y_{\text{cross}}^2} - D_1$, plus the length of the body that is posterior to the CoM $L_{\text{prey}}$). Because a larger $|\alpha|$ requires further turning prior to forward locomotion, which takes time (*Domenici and Blake, 1991*; *Ellerby and Altringham, 2001*), and the initial velocity after turning was dependent on $|\alpha|$ in our experiment (see Figure 4B), $T_1$ is given by a function of $|\alpha|$ [ $T_1(|\alpha|)$ ]. Therefore, $T_{\text{prey}}$ can be expressed as:

$$T_{\text{prey}} = T_1(|\alpha|) + \frac{\sqrt{X_{\text{cross}}^2 + Y_{\text{cross}}^2} - D_1 + L_{\text{prey}}}{U_{\text{prey}}} \tag{6}$$

$T_{\text{pred}}$ can be expressed as:

$$T_{\text{pred}} = \begin{cases} \frac{D_{\text{initial}} + D_2 - X_{\text{cross}}}{U_{\text{pred}}}, \quad \alpha + \beta < \theta_{\text{corner}} \\ \frac{D_{\text{initial}} + D_{\text{attack}}}{U_{\text{pred}}}, \quad \alpha + \beta \geq \theta_{\text{corner}} \end{cases} \tag{7}$$

where $D_{initial}$ is the distance between the prey and the predator at the onset of the prey's escape response (i.e., the flight initiation distance or reaction distance), and $U_{pred}$ is the predator speed, which is assumed to be constant. From *Equations 5–7*, $T_{diff}$ can be calculated as:

$$T_{\mathbf{diff}} = \begin{cases} \frac{D_{\mathbf{initial}}}{U_{\mathbf{pred}}} + \frac{D_2}{U_{\mathbf{pred}}} - \frac{X_{\mathbf{cross}}}{U_{\mathbf{pred}}} - T_1(|\boldsymbol{\alpha}|) - \frac{\sqrt{X_{\mathbf{cross}}^2 + Y_{\mathbf{cross}}^2}}{U_{\mathbf{prey}}} + \frac{D_1}{U_{\mathbf{prey}}} - \frac{L_{\mathbf{prey}}}{U_{\mathbf{prey}}}, & \alpha + \beta < \theta_{\mathbf{corner}} \\ \frac{D_{\mathbf{initial}}}{U_{\mathbf{pred}}} + \frac{D_{\mathbf{attack}}}{U_{\mathbf{pred}}} - T_1(|\boldsymbol{\alpha}|) - \frac{\sqrt{X_{\mathbf{cross}}^2 + Y_{\mathbf{cross}}^2}}{U_{\mathbf{prey}}} + \frac{D_1}{U_{\mathbf{prey}}} - \frac{L_{\mathbf{prey}}}{U_{\mathbf{prey}}}, & \alpha + \beta \geq \theta_{\mathbf{corner}} \end{cases} \tag{8}$$

Because $\frac{D_{initial}}{U_{pred}} + \frac{D_1}{U_{prey}} - \frac{L_{prey}}{U_{prey}}$ are independent of $\alpha$ and $\beta$, we can calculate the relative values of $T_{diff}$ ($T'_{diff}$) in response to the changes of $\alpha$ and $\beta$, from:

$$T_{\mathbf{diff}}{'} = \begin{cases} \frac{D_2}{U_{\mathbf{pred}}} - \frac{X_{\mathbf{cross}}}{U_{\mathbf{pred}}} - T_1(|\boldsymbol{\alpha}|) - \frac{\sqrt{X_{\mathbf{cross}}^2 + Y_{\mathbf{cross}}^2}}{U_{\mathbf{prey}}}, & \alpha + \beta < \theta_{\mathbf{corner}} \\ \frac{D_{\mathbf{attack}}}{U_{\mathbf{pred}}} - T_1(|\boldsymbol{\alpha}|) - \frac{\sqrt{X_{\mathbf{cross}}^2 + Y_{\mathbf{cross}}^2}}{U_{\mathbf{prey}}}, & \alpha + \beta \geq \theta_{\mathbf{corner}} \end{cases} \tag{9}$$

Because $X_{cross}$ and $Y_{cross}$ are dependent on $D_{width}$, $D_{attack}$, and $R_{device}$ as well as $\alpha + \beta$, and $D_2$ is dependent on $D_{width}$ and $R_{device}$, we can calculate $T'_{diff}$ in response to the changes of $\alpha$ and $\beta$, from $D_1$, $D_{width}$, $D_{attack}$, $R_{device}$, $U_{prey}$, $U_{pred}$, and $T_1(|\alpha|)$. Given that the escape success is assumed to be dependent on $T'_{diff}$, the theoretically optimal ET can be expressed as:

$$\text{The optimal ET} = \underset{\boldsymbol{\alpha} + \boldsymbol{\beta}}{\mathrm{argmax}}\left(T_{\mathbf{diff}}{'}\right) \tag{10}$$

## Results

### Experimental results

*Pagrus major* exhibited a typical C-start escape response (*Figure 2—figure supplement 2*; *Figure 3—figure supplement 1*), which consists of the initial bend (stage 1), followed by the return tail flip (stage 2), and continuous swimming or coasting (stage 3) (*Domenici and Blake, 1997*; *Weihs, 1973*). *Figure 3* shows the effect of the initial orientation $\beta$ on the ETs. As was done in previous studies (*Domenici et al., 2011b*; *Domenici et al., 2009*; *Nair et al., 2017*), the away (contralateral) and toward (ipsilateral) responses, defined as the first detectable movement of the fish oriented either away from or toward the predator, were analyzed separately. When the initial orientation was small (i.e., the prey was attacked head-on; *Figure 3A*; 0°≤β<30°), two peaks in the ET distribution were observed: a larger peak at around 100° (away response) and a smaller one at around −80° (toward response). As the initial orientation increases (*Figure 3A*; 30°≤β<60°), the peak at around −80° disappeared. As the initial orientation further increases beyond 60°, another peak appeared at around 170° (*Figure 3A*). When the initial orientation was large (i.e., the prey was attacked from behind; *Figure 3A*; 150°≤β≤180°), there were two similar-sized peaks in the ET at around 130° (toward response), and 180–200° (away response). There were significant effects of initial orientation on the ET in both the away and the toward responses (away: generalized additive mixed model [GAMM] $F$=214.81, p<0.01, n=208; toward: GAMM, $F$=373.92, p<0.01, n=56). There were significant effects of initial orientation on the turn angle $\alpha$ in away and toward responses (*Figure 3—figure supplement 2*; away: GAMM, $F$=90.88, p<0.01, n=208; toward: GAMM, $F$=42.48, p<0.01, n=56). In the overall frequency distribution of ETs pooling the data on all initial orientations and both toward and away responses, there were two large peaks at 120–130° and 170–180°, and one small peak at around −80° (*Figure 3C*). These three peaks were confirmed by the Gaussian mixture model analysis (*Domenici et al., 2008*), where we fitted one to nine Gaussian curves to the ETs, and selected the most parsimonious model based on the Akaike information criterion (AIC) (*Figure 3—source data 1*).

There were no significant effects of predator speed on the ET and $|\alpha|$ in either the toward or the away responses (ET, away: GAMM, $F$=0.01, p=0.93, n=208; ET, toward: GAMM, $F$=0.05, P=0.82, n=56; $|\alpha|$, away: GAMM, $F$=0.01, p=0.93, n=208; $|\alpha|$, toward: GAMM, $F$=0.05, p=0.82, n=56). There were no significant effects of predator speed (slow [from the minimum to the 33.3% quantile]: 0.13–0.93 m s$^{-1}$; and fast [from the 66.7% quantile to the maximum]: 1.29–1.88 m s$^{-1}$) on the variations of ETs and $|\alpha|$ in all 30° initial orientation bins (Levene's test, W=0.02–3.22, p=0.09–0.88, n=22–47).

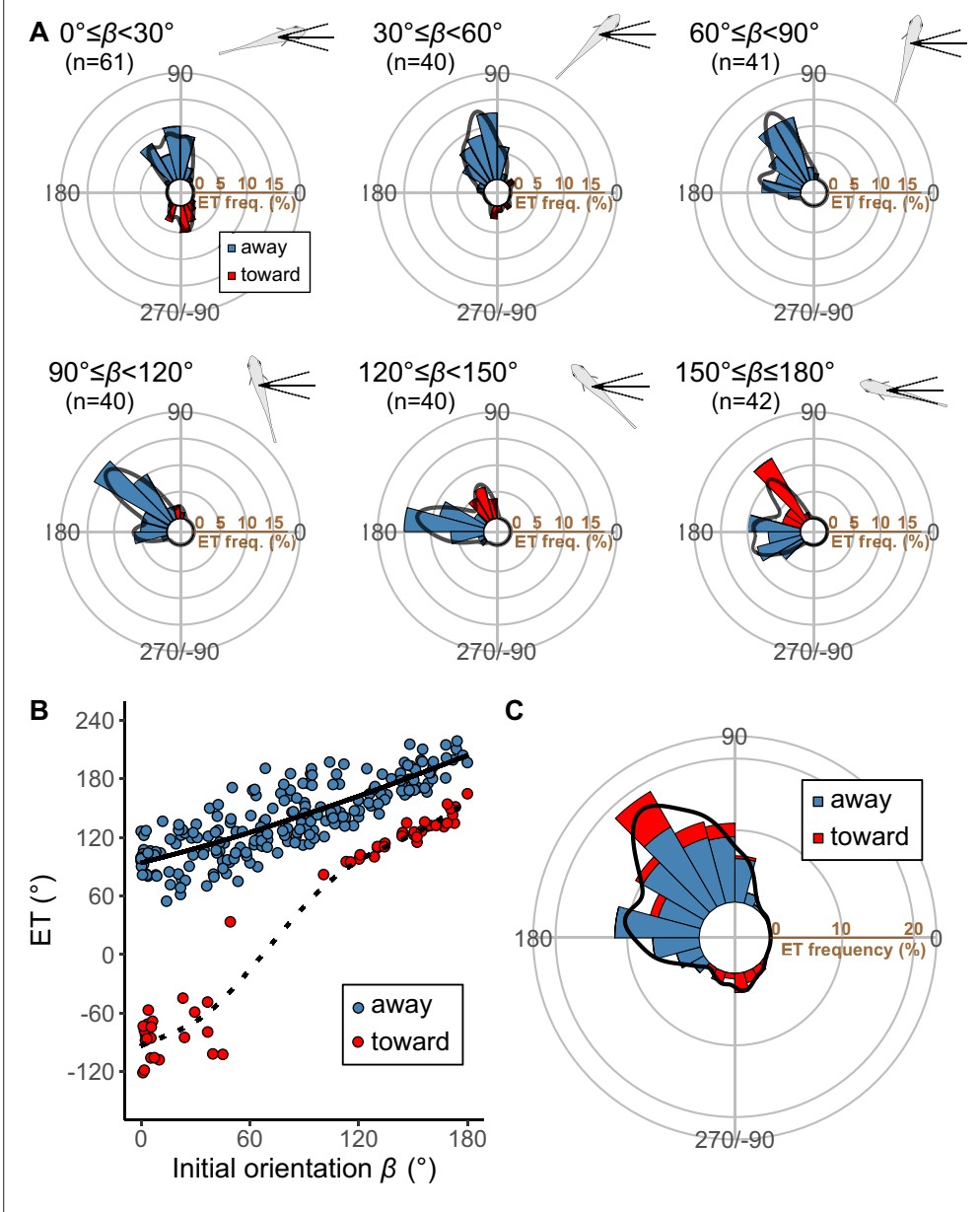

**Figure 3.** Results of the experiments of *Pagrus major* attacked by a dummy predator (i.e., a cast of *Sebastiscus marmoratus*). (**A**) Circular histograms of escape trajectories (ETs) in 30° initial orientation *β* bins. Solid lines are estimated by the kernel probability density function. Concentric circles represent 5% of the total sample sizes within each *β* bin, the bin intervals are 15°, and the bandwidths of the kernel are 50. A drawing of the prey and predator's approach direction is shown in the upper-right corner of each graph. The arrow and dotted lines represent the median value and range of predator's approach direction, respectively. (**B**) Relationship between initial orientation and ET. Different colors represent the away (blue) and toward (red) responses. Solid and dotted lines are estimated by the generalized additive mixed model (GAMM). (**C**) Circular histogram of ETs pooling all the data shown in A. Solid lines are estimated by the kernel probability density function. Concentric circles represent 10% of the total sample sizes, the bin intervals are 15°, and the bandwidths of the kernel are 50. The predator's approach direction is represented by 0°. The dataset and R code are available at Figshare ('Dataset1.csv' and 'Source code 1.R') (*n*=264 [208 away and 56 toward responses] from 23 individuals).

The online version of this article includes the following source data and figure supplement(s) for figure 3:

**Source data 1.** Akaike information criterion for one to nine Gaussian mixture models to estimate the escape trajectory (ET) distribution.

*Figure 3 continued on next page*

*Figure 3 continued*

**Figure supplement 1.** Representatives of kinematic variables of the prey *Pagrus major* and the predator over time.

**Figure supplement 2.** Relationship between initial orientation $\beta$ and turn angle $\alpha$ in the experiment.

**Figure supplement 3.** Experimental apparatus.

## Determination of parameter values

To predict the relationship between the ET ($\alpha+\beta$) and the relative time difference $T_{diff}$ in each initial orientation ($\beta$) by the geometric model, we needed $D_{width}$, $R_{device}$, $D_1$, $U_{prey}$, $T_1(|\alpha|)$, $D_{attack}$, and $U_{pred}$. The methods for determining parameter values are summarized in *Table 1*. $D_{width}$ and $R_{device}$ were determined from the mouth shape of the predator (the sacrificed specimen for making the dummy predator) when fully opened, which were 18 and 199 mm, respectively. $D_1$, $U_{prey}$, and $T_1(|\alpha|)$ were directly estimated by analyzing the escape responses of the prey. Because we have no previous knowledge about the values of $U_{pred}$ and $D_{attack}$ that the prey regards as dangerous, optimal values of $U_{pred}$ and $D_{attack}$ were determined iteratively by comparing model outputs with observed ETs. These optimal values were checked afterward with the data from previously published predator-prey experiments on the same species of prey and predator (*Kimura and Kawabata, 2018*). We applied this optimization procedure to estimating $U_{pred}$ instead of measuring the dummy predator speed per trial in the experiment because there was no significant effect of predator speed on ET in the experiment, suggesting that the prey is likely to have optimized their ETs based on a fixed predator speed (see Discussion for details). This assumption was also supported by the follow-up analysis using the dummy predator speed per trial, where the model fits became worse compared to the model using the fixed predator speed estimated through the optimization procedure (*Table 3—source data 1*; *Figure 5—figure supplement 1*).

The distance of the fast-start phase ($D_1$) was regarded as 15 mm based on the relationship between displacement and velocity of the prey in the experiments (*Figure 4—figure supplement 1*), where the velocity increased up to about 15 mm of displacement from the initial position, beyond which it plateaus; over the 15 mm displacement from the initial position, there were no significant differences

**Table 1.** Methods for determining parameter values.

| Symbol | Description | Value | Method |
|---|---|---|---|
| $D_{width}$ | The half-width of the predator capture device (e.g., mouth) | 18 mm | Measured directly from the dummy predator (a sacrificed individual) |
| $R_{device}$ | The radius of the predator's capture device at the moment of attack, which is approximated as an arc | 199 mm | Measured directly from the dummy predator (a sacrificed individual) |
| $D_1$ | The displacement from the initial position of prey where it was assumed to end the fast-start phase | 15 mm | Estimated from the escape kinematics of prey in the experiment |
| $U_{prey}$ | The prey speed after the displacement of $D_1$, which is assumed to be constant | 1.04 m s$^{-1}$ | Estimated from the escape kinematics of prey in the experiment |
| $T_1(|\alpha|)$ | The time required for a displacement of $D_1$ from the initial position of the prey, given by a function of turn angle $|\alpha|$ | *Figure 4A* | Estimated from the escape kinematics of prey in the experiment |
| $D_{attack}$ | The distance between the prey's initial position and the tip of the predator capture device at the end of the predator attack | 35 mm | Optimized by comparing the model outputs with experimental data |
| $U_{pred}$ | The predator speed, which is assumed to be constant | 1.54 m s$^{-1}$ | Optimized by comparing the model outputs with experimental data |

**Table 2.** Widely applicable or Watanabe-Akaike information criterion (WAIC) for each model in the hierarchical Bayesian models ($n=263$ and 264, respectively, from 23 individuals).

The dataset and R code are available at Figshare ('Dataset1.csv', 'Source code 2.pdf', and 'Source code 3.pdf').

| Relationship | WAIC | ΔWAIC |
|---|---|---|
| $|\alpha|$–$T_1$ relationship | | |
| Piecewise linear | **1363.7** | **0** |
| Linear | 1376.7 | 7.0 |
| Constant | 1581.1 | 217.4 |
| $|\alpha|$-initial velocity after stage 1 turn relationship | | |
| Piecewise linear | **−218.1** | **0** |
| Linear | −205.1 | 13.0 |
| Constant | −171.5 | 46.6 |

$|\alpha|$, absolute value of the turn angle; $T_1$, time required for a displacement of 15 mm from the initial position. The best models are shown in bold.

The online version of this article includes the following source data for table 2:

**Source data 1.** The case where the distance for the fast-start phase was regarded as either 10 or 20 mm.

in the mean velocity between any combinations of 3 mm intervals in any 30° $|\alpha|$ bins (***Figure 4—figure supplement 1***; paired *t*-test with Bonferroni's correction, all p=1.00, *n*=23). There were significant effects of $|\alpha|$ on the time for a displacement of 15 mm from the initial position (GAMM, *F*=78.84, p<0.01, *n*=263; note that the sample size is smaller than the total number of observations, 264, because the prey did not move over 15 mm in one case) and on the mean velocity during the displacement (GAMM, *F*=76.00, p<0.01, *n*=263). However, there were no significant effects of $|\alpha|$ on the time required for a displacement of 15–30 mm from the initial position (GAMM, *F*=1.52, p=0.22, *n*=257; note that the sample size is smaller than the total number of observations, 264, because the prey did not move over 30 mm in seven cases) and on the mean velocity during the displacement (GAMM, *F*=0.89, p=0.27, *n*=257). Therefore, the time required for the prey to turn was incorporated into the model by analyzing the relationship between $|\alpha|$ and the time required for a displacement of 15 mm. The mean velocity of the prey during the constant phase $U_{prey}$ was estimated to be 1.04 m s⁻¹, based on the experimental data. Because the cut-off distance might affect the overall results of the study, we have repeated all the statistical analyses (see ***Tables 2 and 3***, and the text below for results with a cut-off distance of 15 mm) with cut-off distances of 10 and 20 mm and confirmed that the overall results are insensitive to the changes (***Table 2—source data 1***; ***Table 3—source data 2***).

The relationship between $|\alpha|$ and the time required for a displacement of 15 mm, $T_1(|\alpha|)$, is shown in ***Figure 4***. The time was constant up to 44° of $|\alpha|$, above which the time linearly increased in response to the increase of $|\alpha|$ (***Figure 4A***). In the hierarchical Bayesian model, the lowest widely applicable or Watanabe-Akaike information criterion (WAIC) was obtained for the piecewise linear regression

**Table 3.** Comparison of the distribution of escape trajectories (ETs) between the model prediction ($n=264$ per simulation × 1000 times) and experimental data ($n=264$) using the two-sample Kuiper test.

The dataset and R code are available at Figshare ('Dataset1.csv' and 'Source code 1.R').

| Model | Median Kuiper's *V* | Median p | Rate of p>0.05 |
|---|---|---|---|
| With both $D_{attack}$ and $T_1(|\alpha|)$ | 0.11 | 0.44 | 0.97 |
| With $D_{attack}$ and without $T_1(|\alpha|)$ | 0.26 | <0.01 | 0.00 |
| Without $D_{attack}$ and with $T_1(|\alpha|)$ | 0.18 | <0.01 | 0.12 |
| Neither $D_{attack}$ nor $T_1(|\alpha|)$ | 0.28 | <0.01 | 0.00 |

$D_{attack}$, distance between the prey's initial position and the endpoint of the predator attack; $T_1(|\alpha|)$, relationship between the absolute value of the turn angle and the time required for a 15 mm displacement from the initial position (i.e., the time required for the prey to turn).

The online version of this article includes the following source data for table 3:

**Source data 1.** The case where $U_{pred}$ was determined from the dummy predator speed per trial in the experiment.

**Source data 2.** The case where the distance for the fast-start phase was regarded as either 10 or 20 mm.

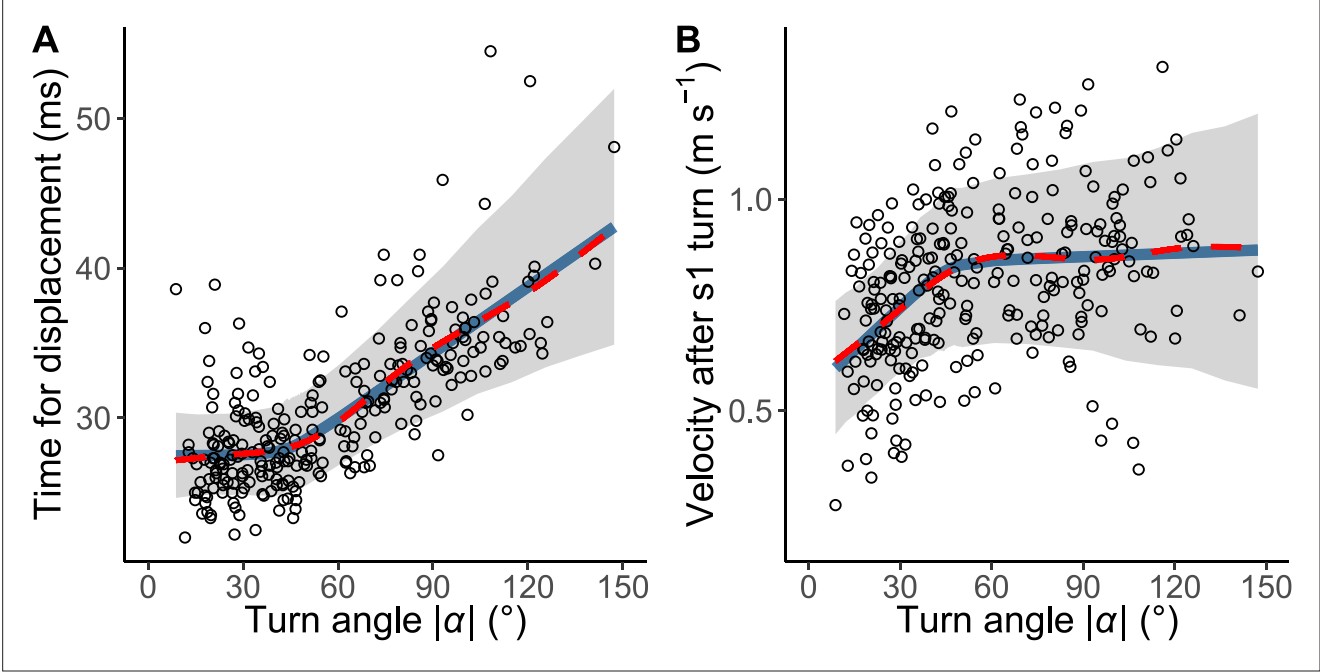

**Figure 4.** The relationship between the absolute value of the turn angle |α| and time-distance variables. (**A**) Relationship between |α| and the time required for a displacement of 15 mm from the initial position of the prey (n=263 from 23 individuals). (**B**) Relationship between |α| and the initial velocity after stage 1 turn (n=264 from 23 individuals). Solid blue lines are estimated by the piecewise linear regression model, and red dashed lines are estimated by the generalized additive mixed model (GAMM). The shaded regions indicate the 95% Bayesian credible intervals of the piecewise linear regression model. The dataset and R code are available at Figshare ('Source code 1.R', 'Source code 2.pdf', 'Source code 3.pdf', and 'Dataset1.csv').

The online version of this article includes the following figure supplement(s) for figure 4:

**Figure supplement 1.** Relationship between displacement from the initial position (3 mm intervals: 0–3, 3–6,..., and 27–30 mm) and mean velocity during the displacement for each turn angle (|α|) bin.

model (***Table 2***). To understand the possible mechanism of the relationship, the relationship between |α| and initial velocity after a stage 1 turn, calculated as the displacement per second during the 10 ms after the turn, was also evaluated (***Figure 4B***). The velocity increased in response to |α| up to 46°, beyond which it plateaus. In the hierarchical Bayesian model, the lowest WAIC was obtained for the piecewise linear regression model (***Table 2***). In both relationships, the regression lines by the piecewise linear model were similar to those by the GAMM, suggesting that the general trends of the relationships were clearly captured by this method. The change points of the two relationships were not significantly different (difference: 1.70±18.01° [mean ± 95% Bayesian credible intervals]). These results indicate that fish with a small |α| (<<45°) can accomplish the stage 1 turn quickly but their velocity after the turn is lower, while fish with an intermediate |α| (=45°) spend a longer time on the stage 1 turn, but their velocity after the turn is higher. Fish with a large |α| (>>45°) spend a still longer time on the stage 1 turn, but their velocity after the turn is similar to that with an intermediate |α| (***Figure 4***).

We have optimized the values of $U_{pred}$ and $D_{attack}$ from the perspective of the prey using the experimental data (see Materials and methods for details). Briefly, the optimal values for prey were obtained using the ranking index, where 0 means that the real fish chose the theoretically optimal ET where $T_{diff}$ is the maximum, and 1 means that the real fish chose the theoretically worst ET where $T_{diff}$ is the minimum (e.g., going toward the predator). The result shows that the optimal value of $D_{attack}$ is 34.73 mm and the optimal value of $U_{pred}$ is 1.54 m s⁻¹. Using data from previously published predator-prey experiments on the same species of prey and predator (***Kimura and Kawabata, 2018***), we show that the estimated $D_{attack}$ value is at the upper limit of the empirical data and the estimated $U_{pred}$ value is higher than the mean of the observed predator speed (***Figure 5—figure supplement 2A, B***). Similarly, the estimated $U_{pred}$ value is higher than the mean of the observed dummy predator speed in our experiment (***Figure 5—figure supplement 2C, D***). These results suggest that the values

independently estimated in the present study are reasonable, and the prey may choose ETs by over-estimating the values of $D_{attack}$ and $U_{pred}$.

## Comparison of model predictions and experimental data

*Figure 5A* plots the relationships between the ET and the relative time difference $T_{diff}$ for different initial orientations $\beta$, estimated by the geometric model; *Figure 5B* plots the relationship between the initial orientation and the theoretical ET. Forty-one percent, 76%, and 94% of observed ETs were within the top 10%, 25%, and 40% quantiles, respectively (0.1, 0.25, 0.40 ranking index) of the theoretical ETs (*Figure 5B* and *Figure 5—figure supplement 3*). In general, the predicted ETs are in line with the observed ones, where the model predicts a multimodal pattern of ET with a higher peak (i.e., optimal ET) at the maximum $T_{diff}$ ($T_{diff,1}$) and a second lower peak (i.e., suboptimal ET) at the second local maximum of $T_{diff}$ ($T_{diff,2}$). When the initial orientation is <20° (*Figure 5A*; $\beta$=15°, *Figures 5B and 6B*), the optimal and suboptimal ETs are around 100° (away response) and −100° (toward response), respectively, which is consistent with the bimodal distribution of our experiment (*Figure 3A*; 0°≤$\beta$<30°). At initial orientations in the range 20–60°, the suboptimal ET switches from around −100° to 170° (*Figure 5A*; $\beta$=45°, *Figures 5B and 6B*), although $T_{diff,2}$ is extremely small compared to $T_{diff,1}$ (*Figure 5A*; $\beta$=45°, *Figures 5B and 6B*). Accordingly, the second peak (i.e., at around 170°) was negligible in our experimental data (*Figure 3A*; 30°≤$\beta$<60°), even though the fish can potentially reach such an ET (i.e., from such an initial orientation, an 170° ET is within the upper limit of |$\alpha$|, 147°). When the initial orientation is 60–120° (*Figure 5A*; $\beta$=75° and $\beta$=105°, *Figures 5B and 6B*), the optimal ET is 100–140° (gradually shifting from 100° to 140°), and the suboptimal ET is around 170°. These two peaks and the shift of the optimal ET are consistent with the experimental results (*Figure 3A*; 60°≤$\beta$<90° and 90°≤$\beta$<120°). The values of the optimal and suboptimal ETs are reversed at initial orientations > 120° (*Figures 5B and 6B*), as the optimal and suboptimal values become 170–180° and around 140°, respectively (*Figure 5A*). These results are again consistent with the bimodal distribution of our experiments (*Figure 3A*; 120°≤$\beta$<150° and 150°≤$\beta$≤180°).

*Figure 5C* shows the circular histogram of the overall theoretical ETs estimated by Monte Carlo simulation. The theoretical ETs show two large peaks at around 110–130° and 170–180°, and one small peak at around −100° (*Figure 5C*). This theoretically estimated ET distribution is similar to the frequency distribution of the observed ETs (*Figure 3C*); there were no significant differences in the frequency distribution between theoretical ETs ($n$=264 per simulation) and observed ETs ($n$=264) in 971 of 1000 simulations (*Table 3*; two-sample Kuiper test, median $V$=0.11, median p=0.44).

To investigate how the initial orientation of the prey modulates the proportion of using the theoretically optimal ET (i.e., where $T_{diff}$ is the maximum, $T_{diff,1}$) compared to using the suboptimal ET (i.e., where $T_{diff}$ is the second local maximum, $T_{diff,2}$), we calculated the optimal ET advantage ($T_{diff,1}-T_{diff,2}$) (*Figure 6A*), which represents the difference in the buffer time available for the prey to escape from the predator, at different initial orientations. The fish chose the optimal and suboptimal ETs to a similar extent when the optimal ET advantage is negligible (*Figure 6C*). For example, when looking at the optimal ET advantage <2 ms, where the initial orientation is 0–7° and 106–180° (46% of all initial orientations), the proportion of the optimal ET used was only 55% (*Figure 6B and C*). On the other hand, the proportion of the optimal ET used was 81% when the optimal ET advantage is higher than 6 ms (i.e., when the initial orientation is 21–75°) (*Figure 6B and C*). There was a significant effect of optimal ET advantage on the proportion of the optimal ET used by fish tested in our experiments (mixed-effects logistic regression analysis, $\chi^2$=10.72, p<0.01, $n$=247).

To investigate the effects of two factors (i.e., the endpoint of the predator attack $D_{attack}$ and the time required for the prey to turn $T_1(|\alpha|)$) on the predictions of ET separately, we constructed three additional geometric models (*Figure 5—figure supplements 4–6*): a model that includes only $D_{attack}$, a model that includes only $T_1(|\alpha|)$, and a null model that includes neither factors (*Figure 2A* and *Domenici, 2002*). In all of these models, the theoretical ET distributions estimated through Monte Carlo simulations were significantly different from the observed ET distributions (*Table 3*; two-sample Kuiper test, median p<0.01). Although the model with $D_{attack}$ and the model with $T_1(|\alpha|)$ show multimodal patterns of ET distribution, the simulation based on these models do not match the experimental data, likely because of differences in the values and relative heights of the peaks (*Figure 5—figure supplements 4 and 5*). The null model shows a unimodal pattern of ET distribution (*Figure 5—figure supplement 6*).

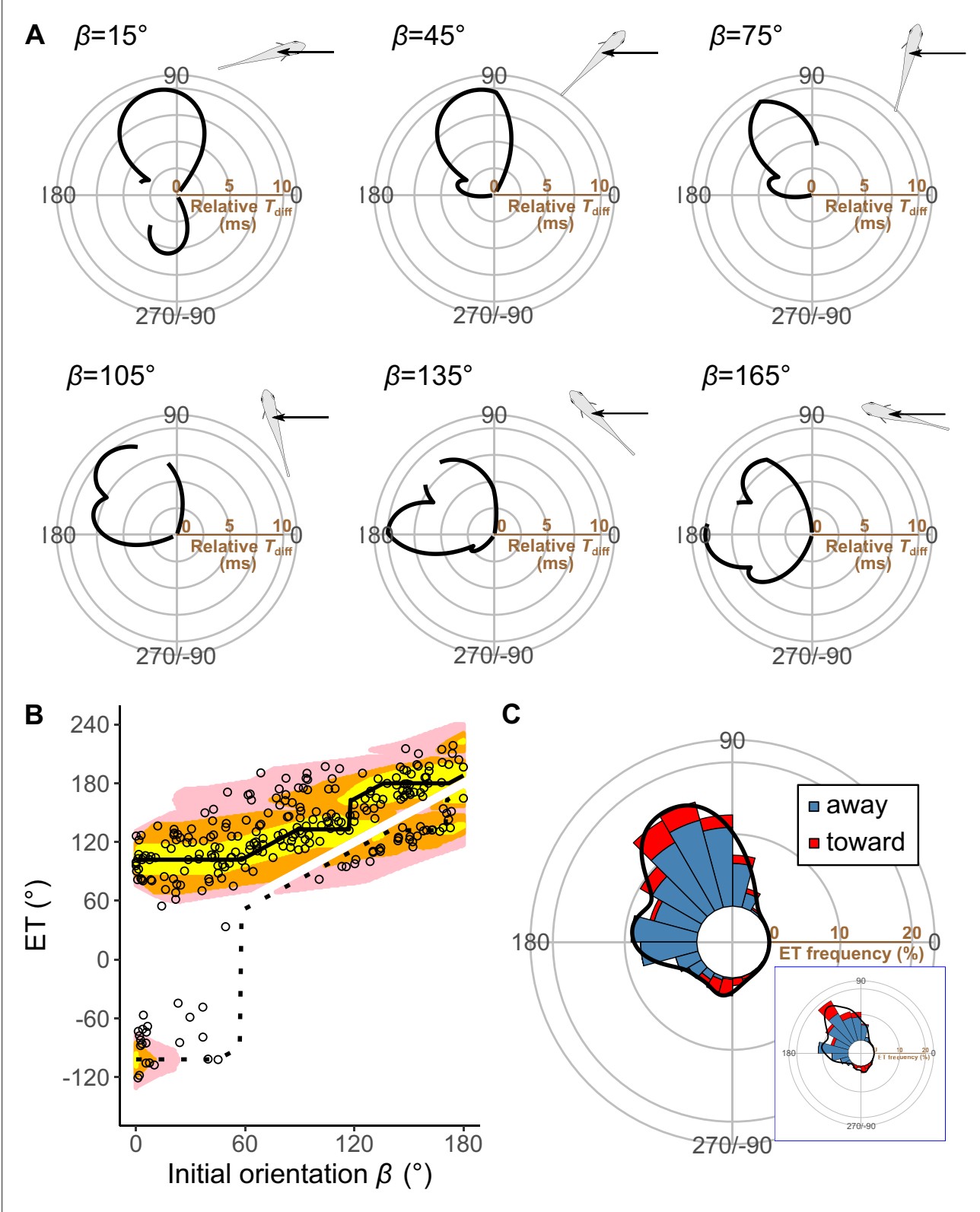

**Figure 5.** Model estimates. (**A**) Relationship between the escape trajectory (ET) and the time difference between the prey and predator $T_{diff}$ in different initial orientations $\beta$. The time difference of the best ET was regarded as 10 ms, and the relative time differences between 0 and 10 ms are shown by solid lines. Areas without solid lines indicate that either the time difference is below 0 or the fish cannot reach that ET because of the constraint on the possible range of turn angles $|\alpha|$. A drawing of prey and predator's approach direction (arrow) is shown in the upper-right corner of each graph.

*Figure 5 continued on next page*

Figure 5 continued

(**B**) Relationship between the initial orientation $\beta$ and ET. Solid and dotted lines represent the best-estimated away and toward responses, respectively. Different colors represent the top 10%, 25%, and 40% quantiles of the time difference between the prey and predator within all possible ETs. (**C**) Circular histogram of the theoretical ETs, estimated by a Monte Carlo simulation. The probability of selection of an ET was determined by the truncated normal distribution of the optimal ranking index (*Figure 5—figure supplement 3*). This process was repeated 1000 times to estimate the frequency distribution of the theoretical ETs. Colors in the bars represent the away (blue) or toward (red) responses. Black lines represent the kernel probability density function. Concentric circles represent 10% of the total sample sizes, the bin intervals are 15°, and the bandwidths of the kernel are 50. Circular histogram of the observed ETs (*Figure 3C*) is shown in the lower-right panel for comparison. The predator's approach direction is represented by 0°. The dataset and R code are available at Figshare ('Dataset1.csv' and 'Source code 1.R') (*n*=264 from 23 individuals for experimental data, and *n*=264,000 for Monte Carlo simulation).

The online version of this article includes the following figure supplement(s) for figure 5:

**Figure supplement 1.** Circular histogram of the theoretical escape trajectories (ETs), estimated by a Monte Carlo simulation of the model that uses the dummy predator speed per trial ((**A**) the predator speed at the onset of escape response of prey; (**B**) the mean predator speed to cover 75% of the flight initiation distance of prey).

**Figure supplement 2.** Predator *Sebastiscus marmoratus* attack parameters.

**Figure supplement 3.** Histogram of the ranking index, where 0 indicates that the real fish chose the theoretically optimal escape trajectory (ET) and 1 indicates that the real fish chose the theoretically worst ET.

**Figure supplement 4.** Estimates of the model with $D_{attack}$ (the distance between the prey's initial position and the endpoint of the predator attack) and without $T_1(|\alpha|)$ (the relationship between the absolute value of the turn angle $|\alpha|$ and the time required for a 15 mm displacement from the initial position, or the time required for prey to turn).

**Figure supplement 5.** Estimates of the model with $T_1(|\alpha|)$ (the relationship between the absolute value of the turn angle $|\alpha|$ and the time required for a 15 mm displacement from the initial position, or the time required for the prey to turn) and without $D_{attack}$ (the distance between the prey's initial position and the endpoint of the predator attack).

**Figure supplement 6.** Estimates of the model that includes neither $D_{attack}$ (the distance between the prey's initial position and the endpoint of the predator attack) nor $T_1(|\alpha|)$ (the relationship between the absolute value of the turn angle $|\alpha|$ and the time required for a 15 mm displacement from the initial position, or the time required for the prey to turn).

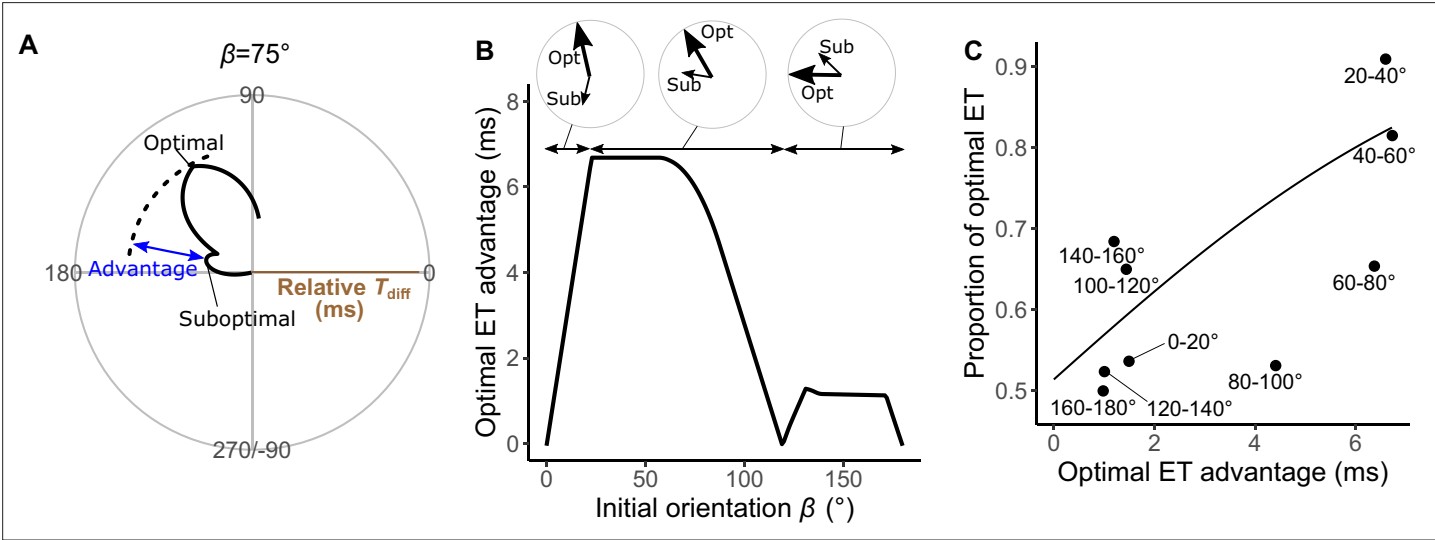

**Figure 6.** Analyses of the probability that the prey chooses the optimal vs. suboptimal escape trajectories (ETs). (**A**) The time difference between the prey and predator $T_{diff}$ at the initial orientation $\beta$ of 75° is shown as an example. We defined the difference between the maximum of $T_{diff}$ (at the optimal ET) and the second local maximum of $T_{diff}$ (at the suboptimal ET) as the optimal ET advantage. (**B**) Relationship between the initial orientation $\beta$ and the optimal ET advantage. Large and small arrows in circles represent the optimal and suboptimal ETs, respectively, for each $\beta$ sectors. (**C**) Relationship between the optimal ET advantage and the proportion of the optimal ET used by the real prey in 20° initial orientation $\beta$ bins. The line was estimated by the mixed-effects logistic regression analysis. The dataset and R code are available at Figshare ('Dataset1.csv' and 'Source code 1.R') (*n*=247 from 23 individuals).

## Potential application of the model to other ET patterns

Although many fish species and animals from other taxa exhibit multiple preferred ETs similar to what we observed here, some animals show different patterns of ETs: for example, a single preferred ET either at around 180° (*Kanou et al., 1999*) or at around 90° (*Cooper, 2006*), and multiple preferred ETs at small and large angles from the predator's approach direction (*Fuiman, 1993*; *Martín and López, 1996*; *Bulbert et al., 2015*; *Figure 7A–C*). To investigate whether our geometric model has the potential to explain these different ET patterns, we changed the values of model parameters (e.g., $U_{pred}$, $D_{attack}$) within a realistic range, and explored whether such adjustments can produce the ET patterns observed in the original work. At small $U_{pred}$, the model predicts one strong peak at around 180° (*Figure 7D*), whereas at large $U_{pred}$, the model predicts a strong peak at around 90° (*Figure 7E*). The model where the predator can adjust the approach path and its attack lasts for a long distance (i.e., large $D_{attack}$) predicts multiple preferred ETs directed at small (at around 30°) and large (at around 170°) angles from the predator's approach direction (*Figure 7F*). These results indicate that our model has the potential to explain various patterns of observed animal ETs. See *Figure 7—figure supplements 1–9* for details of the effect of each parameter on the ET distribution.

## Discussion

Our geometric model, incorporating the endpoint of the predator attack, $D_{attack}$, and the time required for the prey to turn, $T_1(|\alpha|)$, to maximize the difference between the prey and the predator in the time of arrival at the edge of the safety zone, $T_{diff}$, clearly explains the multimodal patterns of ETs in *P. major*. *Figure 8* shows an example of how multiple ETs result in successful escapes from predators. Specifically, according to the model, when the prey escapes at 140° or 170°, it will not be captured by the predator. On the other hand, when the prey escapes along an intermediate trajectory (157°), it will be captured because it swims toward the corner of the danger zone to exit it, and therefore it needs to travel a longer distance than when escaping at 140° or 170°. This example illustrates that the multimodal patterns of ETs are likely to be attributable to the existence of two escape routes: either moving sideways to depart from the predator's strike path or moving opposite to the predator's direction to outrun it. Interestingly, both components of the predator-prey interaction (i.e., $D_{attack}$ and $T_1(|\alpha|)$) added to the previous model (*Domenici, 2002*) are important for accurate predictions of the ET distribution because when they are considered by the model separately, the predictions do not match the experimental data (*Figure 5—figure supplements 4 and 5*; *Table 3*).

Two different escape tactics have been proposed to enhance the success of predator evasion (*Jensen, 2018*; *Domenici et al., 2011a*): the optimal tactic, which maximizes $T_{diff}$ (i.e., the distance between the prey and the predator) (*Arnott et al., 1999*; *Weihs and Webb, 1984*; *Domenici, 2002*; *Soto et al., 2015*), and the protean tactic, which maximizes unpredictability to prevent predators from adjusting their strike trajectories accordingly (*Humphries and Driver, 1970*; *Jones et al., 2011*; *Richardson et al., 2018*; *Moore et al., 2017*). Our results suggest that the prey combines these two different tactics by using multiple preferred ETs. Specifically, when the optimal ET advantage is large (i.e., when the initial orientation is 20–60°), the prey mainly uses the optimal ET (*Figures 3A and 6*). However, when the optimal ET advantage over the suboptimal ET is negligible (i.e., the initial orientation is close to 0° or within the range 110–180°), the prey uses optimal and suboptimal ETs to a similar extent (*Figures 3A and 6*). In such cases, the ET of the prey would be highly unpredictable for the predator. The unpredictability at initial orientations near 0° and 180° is consistent with the study that applied the conventional geometric model to the larval zebrafish *Danio rerio* (*Nair et al., 2017*), where the optimal and suboptimal ETs are approximately symmetrical to the axis of the predator attack. This phenomenon can be explained by the toward-away indecision at orientations nearly perpendicular to the threat (*Domenici and Blake, 1993*; *Domenici and Batty, 1997*). On the other hand, the unpredictability observed at initial orientations near 110–180° is related to the similarly advantageous choice between escaping with an ET at around 140° or 180°. Interestingly, at initial orientations >120°, our results show that these two ETs are reached by using toward and away responses, respectively. The overlap between the ETs of toward and away responses in the overall dataset (*Figure 3*) suggests that toward responses are not 'tactical mistakes' of the prey that turns toward a threat, but are simply related to reaching an optimal or suboptimal ET. These results suggest that the prey strategically adjusts the use of optimal and protean tactics based on their initial orientation. This allows the prey to

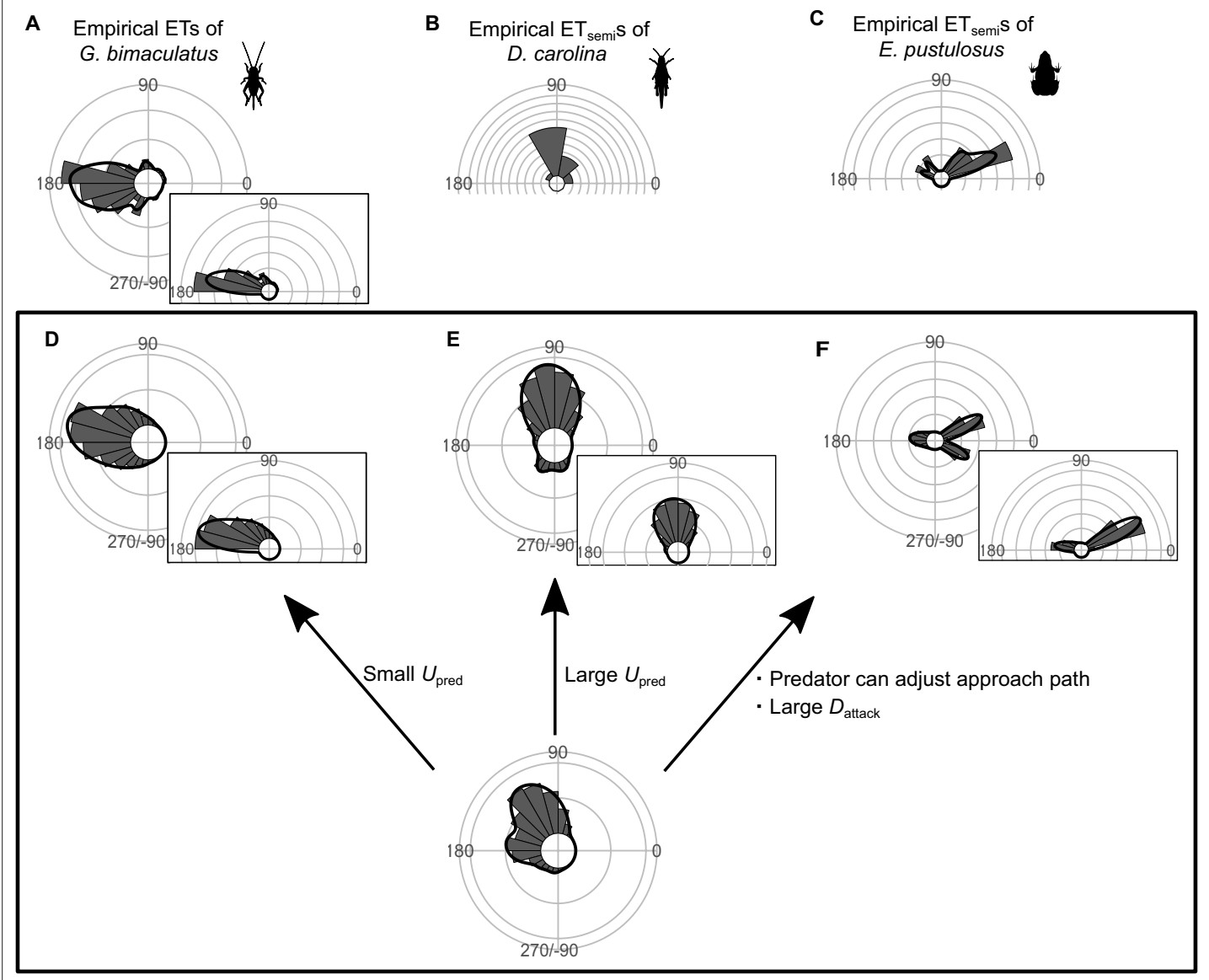

**Figure 7.** Circular histograms of other typical empirical escape trajectory (ET) distribution patterns and the potential explanations by the geometric model. Some previous studies have used the different definition for calculating the angles for ETs, in which the values range from 0° (directly toward the threat) to 180° (opposite to the threat), thereby using only one semicircle regardless of their turning direction and magnitude (e.g., both 120° and 240° of ETs are regarded as 120°). This angle is denoted as $ET_{semi}$, and is shown by a semicircular plot. (**A**) Unimodal ET distribution pattern at around 180° in two-spotted cricket *Gryllus bimaculatus* escaping from the air-puff stimulus. Data were obtained from Figure 4 in ***Kanou et al., 1999***. (**B**) Unimodal $ET_{semi}$ distribution pattern at around 90° in Carolina grasshopper *Dissosteira carolina* escaping from an approaching human. Data were obtained from Figure 3 in ***Cooper, 2006***. (**C**) Bimodal $ET_{semi}$ distribution pattern directed at small and large angles from the predator's approach direction in túngara frog *Engystomops pustulosus* escaping from an approaching dummy bat. Data were obtained from Figure 5b in ***Bulbert et al., 2015*** (**D**) Unimodal ET distribution pattern at around 180°, estimated by a Monte Carlo simulation of the geometric model. In this case, the predator speed $U_{pred}$ is very small (i.e., $K=U_{pred}/U_{prey}$ = 0.3), and the other parameter values are the same as the values used to explain the escape response of *Pagrus major*. (**E**) Unimodal ET distribution pattern at around 90°, estimated by a Monte Carlo simulation of the model. In this case, $U_{pred}$ is very large (i.e., $K=U_{pred}/U_{prey}$ = 7.5), and the other parameter values are the same as the values used to explain the escape response of *P. major*. (**F**) Bimodal ET distribution pattern directed at small and large angles from the predator's approach direction, estimated by a Monte Carlo simulation of the geometric model where the predator can adjust its approach path. In this case, $D_{initial}$ is 130 mm, $D_{react}$ is 70 mm, $R_{turn}$ is 12 mm, $D_{attack}$ is 400 mm, $SD_{choice}$ is 0.23, and the other parameter values are the same as the values used for explaining the escape response of *P. major*. Black lines represent the kernel probability density function with a bandwidth of 50, and concentric circles represent 10% of the total sample sizes. See ***Table 1*** and the text for details of the definitions of the variables. The R code is available at Figshare ('Source code 1.R').

The online version of this article includes the following figure supplement(s) for figure 7:

*Figure 7 continued on next page*

*Figure 7 continued*

**Figure supplement 1.** Effect of predator speed $U_{pred}$ ($K=U_{pred}/U_{prey}$) on the theoretical distribution of escape trajectories (ETs, left panel; $ET_{semi}$, right panel).

**Figure supplement 2.** Effect of $D_{attack}$ (the distance between the prey's initial position and the endpoint of the predator attack) on the theoretical distribution of escape trajectories (ETs, left panel; $ET_{semi}$, right panel).

**Figure supplement 3.** Effect of $R_{device}$ (the radius for the shape of the predator's capture device at the moment of attack, which is approximated as an arc) on the theoretical distribution of escape trajectories (ETs, left panel; $ET_{semi}$, right panel).

**Figure supplement 4.** Effect of $SD_{choice}$ (s.d. of the truncated normal distribution for escape trajectory [ET] choice from the continuum of the optimal ET [ranking index = 0] and worst ET [ranking index = 1]) on the theoretical distribution of ETs (left panel; $ET_{semi}$, middle panel).

**Figure supplement 5.** Effect of predator speed $U_{pred}$ ($K=U_{pred}/U_{prey}$) on the theoretical distribution of escape trajectories (ETs, left panel; $ET_{semi}$, right panel).

**Figure supplement 6.** Effect of $D_{attack}$ (the distance between the prey's initial position and the endpoint of the predator attack) on the theoretical distribution of escape trajectories (ETs, left panel; $ET_{semi}$, right panel).

**Figure supplement 7.** Effect of $D_{initial}$ (the distance between the prey and the predator at the onset of the prey's escape response) on the theoretical distribution of escape trajectories (ETs, left panel; $ET_{semi}$, right panel).

**Figure supplement 8.** Effect of the minimum turning radius of the predator $R_{turn}$ on the theoretical distribution of escape trajectories (ETs, left panel; $ET_{semi}$, right panel).

**Figure supplement 9.** Effect of $SD_{choice}$ (s.d. of the truncated normal distribution for escape trajectory [ET] choice from the continuum of the optimal ET [ranking index = 0] and worst ET [ranking index = 1]) on the theoretical distribution of ETs (left panel; $ET_{semi}$, middle panel).

have unpredictable ETs, thereby preventing predators from anticipating their escape behavior, while keeping $T_{diff}$ large enough to enter the safety zone before the predator reaches it.

From a behavioral decision-making perspective, our results suggest that the prey follows the matching law (*Reed and Kaplan, 2011*; *Poling et al., 2011*; *McDowell, 2013*; *Houston et al., 2021*), where the probability that an optimal or suboptimal ET is chosen is proportional to its chances of survival (i.e., $T_{diff}$). As the matching law predicts (*Houston et al., 2021*), the prey stochastically draws from a Bernoulli distribution dictated by the optimal ET advantage for the binary choice between an optimal and a suboptimal ET, thereby introducing an element of unpredictability, which can prevent predators from learning. Because most empirical studies supporting the matching law use unnatural reinforcement learning paradigms or human behaviors (*Reed and Kaplan, 2011*; *Poling et al., 2011*; *McDowell, 2013*; *Houston et al., 2021*), this result suggests that the matching law is also applicable to animal behavior in realistic contexts. Further research using a real predator and dummy prey (e.g., *Szopa-Comley and Ioannou, 2022*) controlled to escape toward an optimal or suboptimal ET with various specific probabilities is required to test whether our model accurately predicts the best combination of the optimal and suboptimal ETs when accounting for the predator learning.

A relevant question from a perspective of neurosensory physiology is how the animals are able to determine their ETs within milliseconds of response time. The initial orientation of the prey has been incorporated into various neural circuit models (*Eaton et al., 2001*; *Yono and Shimozawa, 2008*; *Card, 2012*; *Levi and Camhi, 2000*), but these models assume that prey animals always escape in a 180° direction (i.e., opposite to the stimulus source), irrespective of the initial orientation. However, the present study shows that animals use suboptimal ETs as well as optimal ETs, and that these ETs may change in a nonlinear fashion, depending on the initial orientation. More specifically, the Mauthner cell and other neurons involved may be activated in accordance with the Bernoulli probabilities dictated by the model, which determine the proportions of away and toward responses and the magnitude of turn to achieve the multiple preferred ETs. Thus, we require new neurophysiological models of ETs to understand how neural circuits process the sensory cues of a threatening stimulus, resulting in muscle actions that generate multiple preferred ETs.

Our geometric model assumes that the prey determines the ETs based on a fixed predator speed. This assumption is supported by the results of our experiments, where the effects of predator speed on the mean and variability of ETs are not significant. Although we did not find any effect of predator speed, it is possible that a speed outside the range we used may affect ETs. Recent studies show that larval zebrafish exhibit less variable ETs under faster threats than they do under slower threats (*Stewart et al., 2014*; *Bhattacharyya et al., 2017*), and the difference in ET variability between fast and slow threats is dependent on whether the Mauthner cell is active or not (*Bhattacharyya et al.,*

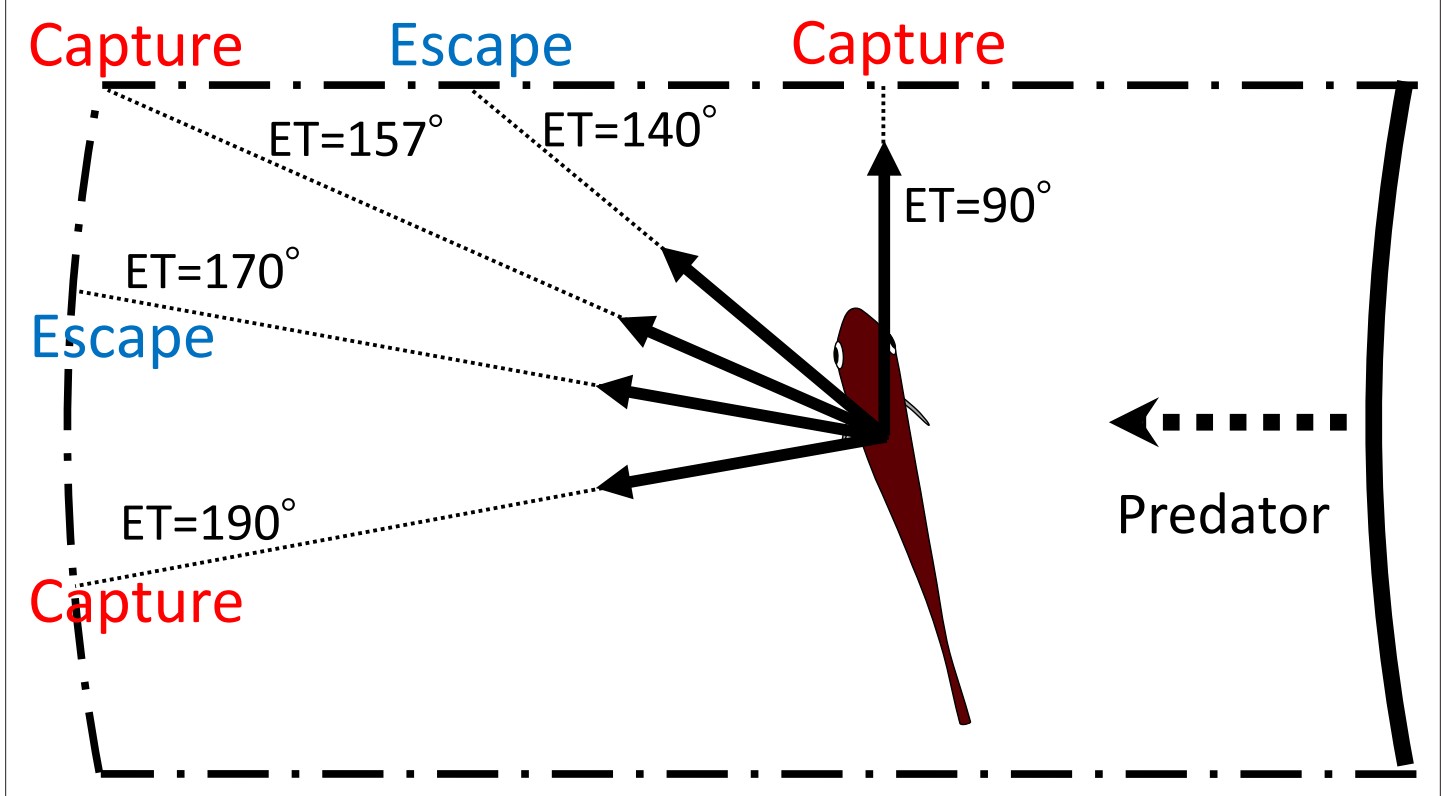

**Figure 8.** Schematic drawing showing how multiple escape trajectories (ETs) result in successful escapes from predators. The area enclosed by dash-dotted lines represents the danger zone the prey needs to exit in order to escape predation, outside of which is the safety zone. When the prey escapes toward the corner of the danger zone (ET = 157°) to exit it, it needs to travel a relatively long distance and therefore the predator can catch it. On the other hand, when the prey escapes with an ET at 170° or 140°, it covers a shorter distance and can reach the safety zone before the predator's arrival. When the prey escapes with an even smaller ET (90°), it will be captured because the shorter travel distance for the predator overrides the benefits of the smaller turn and shorter travel distance for the prey. When the prey escapes with an even larger ET (190°), it will also be captured, because the prey requires a longer time to turn than if escaping along the 170° ET, whereas the travel distance for both predator and prey is the same as that for the 170° ET. In this example, the initial orientation, flight initiation distance, and the body length posterior to the center of mass were set as 110°, 60 mm and 30 mm, respectively.

2017). Therefore, any differences in the ET variability of the present study compared to previous studies could be related to the different involvement of the Mauthner cells. Using the conventional geometric model (*Weihs and Webb, 1984*), Soto et al. showed that the choice of ET only matters to a prey when the predator speed is intermediate, because a prey that is much faster than its predator can escape by a broad range of ETs, whereas a prey that is much slower than its predator cannot escape by any ETs (*Soto et al., 2015*). The predator speed used in this study is in the range of the real predator speed in the previous study using the same species of both predator and prey (*Kimura and Kawabata, 2018*). Thus, our results are ecologically relevant, and the prey is likely to have optimized their ETs based on a fixed predator speed, where the choice of ET strongly affects their survival.

The relationship between $|\alpha|$ and the time required for a 15 mm displacement, $T_1(|\alpha|)$, (*Figure 4A*) indicates that the time required for a 15 mm displacement is relatively constant up to an $|\alpha|$ of about 45°, while a further change in $|\alpha|$ requires additional time. This relationship is likely to be attributable to the kinematics and hydrodynamics of the C-start escape response, because the initial velocity after the stage 1 turn increases linearly up to about 45°, beyond which it plateaus (*Figure 4B*). Interestingly, a recent study on swimming efficiency during acceleration found that efficiency increases linearly with yaw amplitudes up to a certain value, beyond which efficiency plateaus (*Akanyeti et al., 2017*).

Based on the STRANGE framework for animal behavior research (*Webster and Rutz, 2020*), we identified potential biases that may limit the generalizability of our findings. Our empirical data are obtained from one species of hatchery-reared fish with a specific life stage, which has never experienced predators. Therefore, this study alone cannot exclude the possibility that fish of different

species, origins, life stages, and rearing histories have different rules for ETs, which our model cannot explain. However, similar multiple preferred ETs have been observed in many fish species and other animal taxa, including hatcheries/wild origins and different life stages (*Domenici et al., 2011b*). Therefore, we believe that our model is not specific to our experiment but is applicable to other cases showing multiple preferred ETs.

We show that our model has the potential to explain other empirically observed ET patterns (*Figure 7*). Based on the model assuming that the predator makes an in-line attack toward the prey, which is realistic for ambush and stalk-and-attack predators (*Moore and Biewener, 2015*) (e.g., frogs [*Camhi et al., 1978*], spiders [*Dangles et al., 2006*], and fish [*Kimura and Kawabata, 2018*; *Webb and Skadsen, 1980*; *Fouts and Nelson, 1999*; *Rand and Lauder, 1981*]), either single or multiple ETs at around 90–150° and around 180° are predicted, as have been observed in many empirical studies of animals escaping from ambush predators and artificial stimuli (*Domenici et al., 2011b*). Based on the model assuming that the predator can adjust its approach path, which is realistic for pursuit predators, multiple ETs directed at small and large angles from the predator's approach direction can be predicted, as observed in the empirical studies of prey escaping from pursuit predators (*Corcoran and Conner, 2016*; *Bulbert et al., 2015*). Further research measuring the escape response in various species and applying the data to our geometric model is required to verify the applicability of our geometric model to various predator-prey systems.

Our work represents a major advancement in understanding the basis of the variability in ETs observed in previous works (reviewed in *Domenici et al., 2011b*). Our results suggest that prey use multiple preferred ETs to maximize the time difference between themselves and the attacking predator, while keeping a high level of unpredictability. The results also suggest that prey strategically adjust the use of protean and optimal tactics with respect to the advantage of the optimal ET over the suboptimal ET. Because multimodal ETs similar to what we observed here have been found in many fish species and other animal taxa (*Domenici et al., 2011b*), this behavioral phenotype may result from convergent evolution in phylogenetically distant animals. From a neurosensory perspective, our findings open new avenues to investigate how the animals determine their ETs from multiple options with specific probabilities, which are modulated by the initial orientation with respect to the threat.

## Materials and methods
### Definition of the angles
The C-start escape response consists of an initial bend (stage 1), followed by a return tail flip (stage 2), and continuous swimming or coasting (stage 3) (*Domenici and Blake, 1997*; *Weihs, 1973*). In line with previous studies (*Domenici et al., 2011b*; *Nair et al., 2017*; *Stewart et al., 2013*), we defined initial orientation $\beta$, directionality (away or toward responses), turn angle $\alpha$, and ET $\alpha+\beta$ as follows (*Figure 2—figure supplement 1*). *Initial orientation* ($\beta$): the angle between the line passing through the prey's CoM (located at 34% of the total length from the tip of the snout; *Kimura and Kawabata, 2018*) and the tip of the snout at the onset of stage 1, and the midline of the predator model attacking in a straight line. Initial orientation ranges from 0° (i.e., when the prey is attacked from front) to 180° (i.e., when the prey is attacked from behind). *Directionality*: the away and toward responses were defined by the first detectable movement of the fish in a direction either away from or toward the predator, respectively (*Domenici et al., 2011b*). In rare cases ($n$=3; 1.1% of the total observations) where the initial orientation is exactly 0° ($n$=1) or 180° ($n$=2), the counterclockwise and clockwise turns were regarded as away and toward responses, respectively. *Turn angle* ($\alpha$): the angle between the line passing through the CoM and the tip of the snout at the onset of stage 1, and the line passing through the CoM at the onset of stage 1 and the CoM at the end of stage 2. The angles of the away and toward responses are assigned positive and negative values, respectively. *ET* ($\alpha+\beta$): the angular sum of the initial orientation ($\beta$) and the turn angle ($\alpha$). Because the experimental data exhibited no asymmetry in directionality (Fisher's exact test, p=1.00, $n$=264) and ET distribution (two-sample Kuiper test, $V$=0.14, p=0.61, $n$=264), we pooled the ETs from the left and right sides, treating all fish as though they were attacked from the right side (*Domenici et al., 2011b*). ET is a circular variable with a cycle of 360°. As the range of |$\beta$| is 0–180° and the range of |$\alpha$| was 9–147° in the experiment, the ET value can potentially range from −147° to 327°. Circular graphs are shown with angles from 0°

to 360° (*Batschelet, 1981*), where negative values such as −90° correspond to positive values shifted by one cycle (in this case, −90°+360°=270°).

## Experiment

Following the STRANGE framework for animal behavior research (*Webster and Rutz, 2020*), we provide details of the test samples and experimental procedure in the following two subsections.

### Sample fish

We used young-of-year juvenile hatchery-reared red sea bream *P. major* for the experiment. Sixty-five individuals were purchased from commercial hatcheries (Marua Suisan Co., Ltd., Ehime, Japan), where they were reared communally in artificial tanks. After arriving at the laboratory at Nagasaki University, they were kept in a 200 l polycarbonate tank and were fed with commercial pellets (Otohime C2; Marubeni Nisshin Feed Co. Ltd, Tokyo, Japan) twice a day. The sex of the fish was not determined because the species of this size is in a bisexual juvenile stage (*Law and Sadovy de Mitcheson, 2017*). Water temperature was maintained at 23.8–24.9°C.

### Experimental procedure

We have elicited the escape response of *P. major* (45.3±3.5 [39.4–51.5] mm total length, 37.2±2.9 [32.3–42.2] mm standard length, 1.6±0.4 [0.9–2.3] g body weight, mean ± s.d. [range], $n=23$) using a dummy predator. The value of Fulton's condition factor (30.64±2.43 [26.10–35.56], mean ± s.d. [range]), calculated by the body weight of the fish divided by the standard length cubed, suggests that all fish were in a good nutritious condition (*Miyajima-taga et al., 2014*; *Kudoh et al., 2002*). The experiment was conducted in a plastic tank (540 × 890 × 200 mm³) filled with seawater to a depth of 80 mm. The water temperature was maintained at 23.8–24.7°C. An individual *P. major* was randomly captured by a hand net from the holding tank, introduced into a PVC pipe (60 mm diameter) set in the center of the experimental tank, and acclimated for 15 min. Because it was not difficult to capture any individual by a hand net, there should be no bias in selecting individuals with specific behavioral types. After the acclimation period, the PVC pipe was slowly removed, and the dummy predator, a cast of *Sebastiscus marmoratus* (164 mm in total length and 36 mm in mouth width), was moved toward the *P. major* for a distance of 200 mm (*Figure 3—figure supplement 3A*). The dummy predator was held in place by a metal pipe anchored to a four-wheel dolly, which is connected to a fixed metal frame via two plastic rubber bands (*Figure 3—figure supplement 3B*). The wheel dolly was drawn back to provide power for the dummy predator to strike toward the prey. Because the previous work shows that *S. marmoratus* attacks *P. major* using a variable speed (1.10±0.65 [0.09–2.31] m s⁻¹, mean ± s.d. [range]) (*Kimura and Kawabata, 2018*), we used various strengths of plastic rubber bands to investigate the effect of predator speed on ET. The fish movements were recorded from above, using a high-speed video camera (HAS-L1; Ditect Co., Tokyo, Japan) at 500 frames s⁻¹. Each individual *P. major* was stimulated from 5 to 23 times with a time interval of at least 15 min, and, in total, 297 trials were conducted. We eliminated 33 trials from the analysis because *P. major* moved away from the striking course of the dummy predator before the stimulation (in 14 trials) and because bubbles obscured the *P. major* image (in 19 trials). The final data analyzed are 5–20 escape responses per individual and, in total, 264 escape responses. The experiments for each *P. major* were accomplished within 1 day to eliminate possible effects of tank transfer, handling, and change of rearing conditions. The number of recordings of an individual *P. major* was different because we could not allocate the same amount of time to the experiment per day due to the experimental schedule and because the numbers of eliminated data are different among individuals. The recorded videos were analyzed frame by frame using Dipp-Motion Pro 2D (Ditect Co.). The CoM and the tip of the mouth of *P. major* and the tip of the predator's mouth were digitized in each frame to calculate all the kinematic variables. The animal care and experimental procedures were approved by the Animal Care and Use Committee of the Faculty of Fisheries (Permit No. NF-0002), Nagasaki University in accordance with the Guidelines for Animal Experimentation of the Faculty of Fisheries and the Regulations of the Animal Care and Use Committee, Nagasaki University.

## Statistical analysis

Because our geometric model predicts that the initial orientation $\beta$ and the predator speed $U_{pred}$ affect the ET and turn angle $\alpha$, we examined these effects by the experimental data using a GAMM with a normal distribution and identity link function (*Zuur et al., 2009*). ET and $\alpha$ were regarded as objective variables, while predator speed and initial orientation were regarded as explanatory variables and were modeled with a B-spline smoother. Fish ID was regarded as a random factor. Smoothed terms were fitted using penalized regression splines, and the amount of smoothing was determined using the restricted maximum likelihood method. As was done in previous studies (*Domenici et al., 2011b*; *Domenici et al., 2009*; *Nair et al., 2017*), the away and toward responses were analyzed separately. The significance of the initial orientation and predator speed was assessed by the *F*-test. The analysis was conducted using R 3.5.3 (R Foundation for Statistical Computing) with the R package *gamm4*.

## Determination of parameter values

### Determination of the Prey's kinematic parameters

The relationship between $|\alpha|$ and the time required for a displacement of 15 mm, $T_1(|\alpha|)$, was estimated by piecewise linear regression (*Brilleman et al., 2017*). We used piecewise linear regression rather than a commonly used smoothing method such as GAMM, because the smoothing method does not output the timing of the regression change and thus the biological interpretation of the regression curve is problematic (*Brilleman et al., 2017*). The time required for a displacement of 15 mm was regarded as an objective variable, whereas $|\alpha|$ was regarded as an explanatory variable. Fish ID was included as a covariate in order to take into account potential individual differences in the relationship, $T_1(|\alpha|)$. To detect the possible kinematic mechanism of the relationship $T_1(|\alpha|)$, we also examined the relationship between $|\alpha|$ and initial velocity after the stage 1 turn, using piecewise linear regression. Initial velocity after the stage 1 turn was regarded as an objective variable, $|\alpha|$ was regarded as an explanatory variable, and fish ID was included as a covariate. A hierarchical Bayesian model with a Markov chain Monte Carlo (MCMC) method was used to estimate these relationships (*Brilleman et al., 2017*; *Kéry and Schaub, 2011*). The number of draws per chain, thinning rate, burn-in length, and number of chains were set as 200,000, 1, 100,000, and 5, respectively. To test the overall fit of the model, the WAIC of the model was compared with those of the null model (constant) and a simple linear regression model. MCMC was conducted using RStan 2.18.2 (Stan Development Team 2019).

### Determination of predator speed and endpoint of the predator attack

Because we had no previous knowledge about the values of $U_{pred}$ and $D_{attack}$ that the prey regards as dangerous (i.e., the values of $U_{pred}$ and $D_{attack}$ that trigger a response in the prey), we optimized the values using the experimental data in this study. We have input the obtained values of $D_{width}$, $R_{device}$, $D_1$, $U_{prey}$, and $T_1(|\alpha|)$ into the theoretical model. The optimal values of $U_{pred}$ and $D_{attack}$ were obtained using the ranking index. The ranks of the observed ETs among the theoretical ET choices of 1° increment were standardized as the ranking index, where 0 means that the real fish chose the theoretically optimal ET where $T_{diff}$ is the maximum, and 1 means that the real fish chose the theoretically worst ET where $T_{diff}$ is the minimum. The optimal set of $D_{attack}$ and $U_{pred}$ values was determined by minimizing the mean ranking index of the observed ETs. The distribution of the optimal ranking index was then fitted to the truncated normal distribution and was used to predict how the fish chose the ETs from the continuum of the theoretically optimal and worst ETs.

## Model predictions

We input the above parameters ($D_{width}$, $R_{device}$, $D_1$, $U_{prey}$, $T_1(|\alpha|)$, $D_{attack}$, and $U_{pred}$) into the model and calculated how the choice of different ETs affects $T_{diff}$ for each initial orientation $\beta$. Because there was a constraint on the possible range of $|\alpha|$ (i.e., fish escaping by C-start have a minimum and maximum $|\alpha|$ *Domenici and Blake, 1991*), the range of $|\alpha|$ was determined based on its minimum and maximum values observed in our experiment, which were 9–147°.

To estimate the overall frequency distribution of ETs that include the data on observed initial orientations, we conducted Monte Carlo simulations. In each observed initial orientation, the ET was chosen from the continuum of the theoretically optimal and worst ETs. The probability of the ET selection was determined by the truncated normal distribution of the optimal ranking index (e.g., the fish could choose theoretically good ETs with higher probability than theoretically bad ETs, but the

choice is a continuum based on the truncated normal distribution). This process was repeated 1000 times to robustly estimate the frequency distribution of the theoretical ETs. In each simulation run, the frequency distribution of the theoretical ETs was compared with that of the observed ETs using the two-sample Kuiper test (*Zar, 2010*).

To investigate how the real prey changes the probability that it uses the theoretically optimal ET or suboptimal ET, we regarded the difference between the maximum of $T_{diff}$ (at the optimal ET) and the second local maximum of $T_{diff}$ (at the suboptimal ET) as the optimal ET advantage, and theoretically estimated the values for all initial orientations. We then examined the relationship between the optimal ET advantage and the proportion of the optimal ET the prey actually chose using a mixed-effects logistic regression analysis (*Zuur et al., 2009*). Each observed ET was designated as the optimal (1) or the suboptimal (0) based on whether the observed ET was closer to the optimal ET or suboptimal ET. When the prey chose the ET that was more than 35° different from both the optimal and suboptimal ETs, the ET data point was removed from the analysis (these cases were rare: 7%). The choice of ET (optimal (1) or suboptimal (0)) was regarded as an objective variable, while the optimal ET advantage was regarded as an explanatory variable. Fish ID was regarded as a random factor. The significance of the optimal ET advantage was assessed by the likelihood ratio test with $\chi^2$ distribution. The analysis was conducted using R 3.5.3 with the R package *lme4*.

To investigate the effects of two factors (i.e., the endpoint of the predator attack $D_{attack}$ and the time required for the prey to turn $T_1(|\alpha|)$) on predictions of ET separately, we compared four geometric models: the model that includes both $D_{attack}$ and $T_1(|\alpha|)$, the model that includes only $D_{attack}$, the model that includes only $T_1(|\alpha|)$, and the null model. Note that the null model is equivalent to the previous Domenici's model (*Domenici, 2002*). In all models, the values of $U_{pred}$ and $D_{attack}$ were optimized using the ranking index. The overall frequency distributions of ETs were estimated through Monte Carlo simulations, and in each simulation run, the theoretical ET distribution was compared with the observed ET distribution using the two-sample Kuiper test.

To investigate whether our model has the potential to explain other empirical ET patterns, we changed the values of model parameters (e.g., $U_{pred}$, $D_{attack}$) within a realistic range, and conducted Monte Carlo simulations to estimate the frequency distribution of the theoretical ETs. For each initial orientation, ranging from 0° to 180° with an increment of 1°, the ET was chosen based on the probability of the truncated normal distribution (i.e., the continuum of the theoretically optimal and worst ETs), and this process was repeated 100 times. In the model where the predator cannot adjust the strike path (*Figure 2B*), we fixed three parameters and varied the fourth parameter ($U_{pred}$, $D_{attack}$, $R_{device}$, and s.d. of the truncated normal distribution for ET choice, $SD_{choice}$) from the model produced for the escape response of *P. major* (i.e., $D_{attack}$ = 34.73 mm, $U_{pred}$ = 1.54 m s$^{-1}$, $R_{device}$ = 199 mm, $SD_{choice}$ = 0.33). Using the model where the predator can adjust the strike path (*Appendix 1—figure 1B*), we simulate the situation in which the safety zone shape inside the predator's turning radius is similar to the Corcoran's model (*Appendix 1—figure 1A*) but we included a safety zone opposite to the incoming direction of the predator. We considered $D_{attack}$ as 400 mm, $D_{initial}$ as 130 mm, the minimum turning radius of the predator $R_{turn}$ as 12 mm, and the reaction distance of the predator $D_{react}$ as 70 mm. We used the same values of the *P. major* model for $R_{device}$ and the other parameters. We then fixed four parameters and varied the fifth parameter ($U_{pred}$, $D_{attack}$, $D_{initial}$, $R_{turn}$, $SD_{choice}$) to examine the effect of each parameter on the ET distribution.

## Acknowledgements

We sincerely thank YY Watanabe for his constructive comments on an early version of this paper. We appreciate the valuable comments from AD Bolton, C Rutz, and an anonymous reviewer, which significantly improved the manuscript. We also thank H Kamihata for providing equipment for rearing *P. major*. This study was funded by Grants-in-Aid for Scientific Research, Japan Society for the Promotion of Science, to YK (17K17949 and 19H04936), Sumitomo Foundation to YK (153128), and the ISM Cooperative Research Program to YK and KS (2014-ISM.CRP-2006).

# Additional information

## Funding

| Funder | Grant reference number | Author |
| --- | --- | --- |
| Japan Society for the Promotion of Science | Grants-in-Aid for Young Scientists B: 17K17949 | Yuuki Kawabata |
| Sumitomo Foundation | Grant for Environmental Research Projects: 153128 | Yuuki Kawabata |
| ISM Cooperative Research Program | 2014-ISM.CRP-2006 | Yuuki Kawabata Ken-ichiro Shimatani |
| Japan Society for the Promotion of Science | Grant-in-Aid for Scientific Research on Innovative Areas: 19H04936 | Yuuki Kawabata |

The funders had no role in study design, data collection and interpretation, or the decision to submit the work for publication.

## Author contributions

Yuuki Kawabata, Conceptualization, Resources, Data curation, Software, Formal analysis, Supervision, Funding acquisition, Validation, Investigation, Visualization, Methodology, Writing - original draft, Project administration, Writing – review and editing; Hideyuki Akada, Investigation; Ken-ichiro Shimatani, Gregory Naoki Nishihara, Formal analysis, Writing – review and editing; Hibiki Kimura, Investigation, Visualization; Nozomi Nishiumi, Investigation, Writing – review and editing; Paolo Domenici, Conceptualization, Formal analysis, Supervision, Writing – review and editing

## Author ORCIDs

Yuuki Kawabata ⓘ http://orcid.org/0000-0001-8267-5199
Hibiki Kimura ⓘ http://orcid.org/0000-0003-3710-2564

## Ethics

Animal experimentation: The animal care and experimental procedures were approved by the Animal Care and Use Committee of the Faculty of Fisheries (Permit No. NF-0002), Nagasaki University in accordance with the Guidelines for Animal Experimentation of the Faculty of Fisheries and the Regulations of the Animal Care and Use Committee, Nagasaki University.

## Decision letter and Author response

Decision letter https://doi.org/10.7554/eLife.77699.sa1
Author response https://doi.org/10.7554/eLife.77699.sa2

# Additional files

## Supplementary files

• Transparent reporting form

## Data availability

The datasets (Dataset1-5) of the escape response in *P. major*, used for statistical analysis and figures, and the R code (Source code 1-3) for the mathematical model, statistical analysis, and figures are available in Figshare: https://doi.org/10.6084/m9.figshare.17021930.v1.

The following dataset was generated:

| Author(s) | Year | Dataset title | Dataset URL | Database and Identifier |
|---|---|---|---|---|
| Kawabata Y, Akada H, Shimatani K, Nishihara GN, Kimura H, Nishiumi N, Domenici P | 2022 | Datasets and R code for "Multiple preferred escape trajectories are explained by a geometric model incorporating prey's turn and predator attack endpoint" | https://doi.org/10.6084/m9.figshare.17021930.v1 | figshare, 10.6084/m9.figshare.17021930.v1 |

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

## Appendix 1

Mathematical formula for the geometric model modified from *Corcoran and Conner, 2016*. When the prey's CoM at the onset of its escape is located at point (0, 0), the trajectory of the CoM ($X_{prey}$, $Y_{prey}$) is given by:

$$Y_{prey} = X_{prey} \tan (\alpha + \beta) \tag{11}$$

The edge of the safety zone is determined by the half-width of the predator capture device $D_{width}$, the distance between the prey and the predator at the onset of the prey's escape response $D_{initial}$, the distance required for the predator to react the prey's escape response to initiate its turn $D_{react}$, the distance between the prey's initial position and the tip of the predator capture device at the end of the predator attack $D_{attack}$, the minimum turning radius of the predator $R_{turn}$, and the shape of the predator's capture device at the moment of attack, which is approximated as an arc with a certain radius $R_{device}$ (*Appendix 1—figure 1B*). The edge of the safety zone can be divided into four parts: (1) the straight line before the onset of the predator's turn, (2) the arc of the minimum inner turning radius, (3) the capture device shape at the end of the predator attack when it attacks with the minimum turning radius, (4) the involute curve where the tip of the predator capture device traverses a specific distance from the initial position, which can be described by the trace of unwrapping a taut string from the minimum turning circle (*Appendix 1—figure 1B*). Note that the model may lack some parts, depending on the values of parameters.

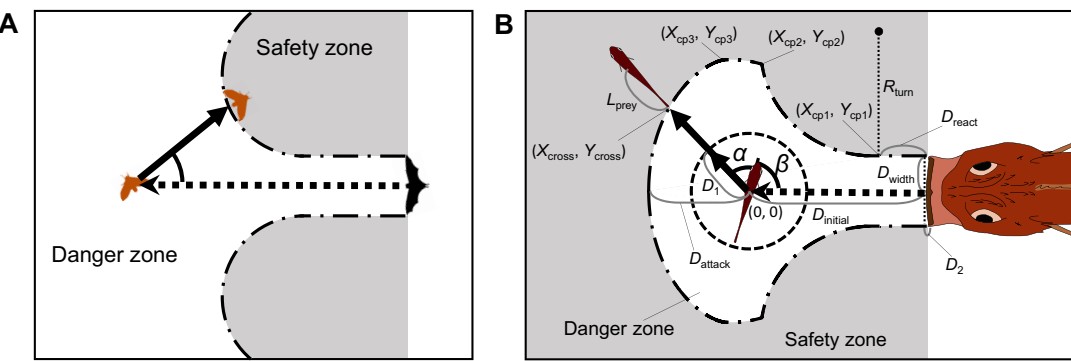

**Appendix 1—figure 1.** Proposed geometric models for animal escape trajectories. (**A**) A previous geometric model proposed by *Corcoran and Conner, 2016*. (**B**) The geometric model modified from *Corcoran and Conner, 2016*. Two factors are added to Corcoran's model: the endpoint of the predator attack, and the time required for the prey to turn. See Appendix 1 for details of the definitions of the variables and mathematical formulas.

The projection of the predator's capture device edge along the X-axis $D_2$ can be expressed as:

$$D_2 = R_{device} \left\{ 1 - \cos \left( \sin^{-1} \frac{D_{width}}{R_{device}} \right) \right\} \tag{12}$$

The angle for the predator to traverse with the minimum turning radius $\gamma$ can be expressed as:

$$\gamma = \frac{D_{initial} - D_{react} + D_{attack}}{R_{turn}} \tag{13}$$

The x and y coordinates of the change point between the first and second parts of the safety zone edge ($X_{cp1}$, $Y_{cp1}$) can be expressed as:

$$\begin{cases} X_{cp1} = D_{initial} - D_{react} + D_2 \\ Y_{cp1} = D_{width} \end{cases} \tag{14}$$

The x and y coordinates of the change point between the second and third parts of the safety zone edge ($X_{cp2}$, $Y_{cp2}$) can be expressed as:

$$\begin{cases} X_{cp2} = D_2\cos\gamma + (D_{width} - R_{turn})\sin\gamma + D_{initial} - D_{react} \\ Y_{cp2} = -D_2\sin\gamma + (D_{width} - R_{turn})\cos\gamma + R_{turn} \end{cases} \tag{15}$$

The x and y coordinates of the change point between the third and fourth parts of the safety zone edge ($X_{cp3}$, $Y_{cp3}$) can be expressed as:

$$\begin{cases} X_{cp3} = -R_{turn}\sin\gamma + D_{initial} - D_{react} \\ Y_{cp3} = -R_{turn}\cos\gamma + R_{turn} \end{cases} \tag{16}$$

The x and y coordinates of the first part of the safety zone edge ($X_{safe1}$, $Y_{safe1}$) are given by:

$$Y_{safe1} = D_{width} \tag{17}$$

The x and y coordinates of the second part of the safety zone edge ($X_{safe2}$, $Y_{safe2}$) are given by:

$$(X_{safe2} - D_{initial} + D_{react})^2 + (Y_{safe2} - R_{turn})^2 = D_2^2 + (D_{width} - R_{turn})^2 \tag{18}$$

The x and y coordinates of the third part of the safety zone edge ($X_{safe3}$, $Y_{safe3}$) are given by:

$$(X_{safe3} - D_{initial} + D_{react} - R_{device}\cos\gamma + R_{turn}\sin\gamma)^2 + (Y_{safe3} - R_{turn} + R_{device}\sin\gamma + R_{turn}\cos\gamma)^2 = R_{device}^2 \tag{19}$$

For calculating the x and y coordinates of the fourth part of the safety zone edge ($X_{safe4}$, $Y_{safe4}$), the formula of involute curve from a circle of radius $R_{turn}$ whose center is the origin (0, 0) with a tip of the string at ($R_{turn}$, 0) is introduced as:

$$\begin{cases} x = R_{turn}\ (\cos\theta + \theta\sin\theta) \\ y = R_{turn}\ (\sin\theta - \theta\cos\theta) \end{cases}, 0 \leq \theta \leq \gamma \tag{20}$$

where $\theta$ denotes an angle for an unwrapped string from the circle in a counterclockwise direction. By moving and rotating this point (x, y), we can calculate the fourth part of the safety zone edge ($X_{safe4}$, $Y_{safe4}$) as:

$$\begin{cases} X_{safe4} = D_{initial} - D_{react} - x\sin\gamma + y\cos\gamma \\ Y_{safe4} = R_{turn} - x\cos\gamma + y\sin\gamma \end{cases} \tag{21}$$

From *Equations 1–11*, to the x and y coordinates of the crossing point of the escape path and the safety zone edge ($X_{cross}$, $Y_{cross}$) are given by a function of $D_{width}$, $D_{attack}$, $R_{device}$, $D_{initial}$, $D_{react}$, $R_{turn}$, and $\alpha+\beta$.

The prey can escape from the predator when the time required for the prey to enter the safety zone ($T_{prey}$) is shorter than the time required for the predator's capture device to reach that entry point ($T_{pred}$). Therefore, the prey is assumed to maximize the difference between the $T_{pred}$ and $T_{prey}$ ($T_{diff}$). To incorporate the time required for the prey to turn, $T_{prey}$ was divided into two phases: the fast-start phase, which includes the time for turning and acceleration ($T_1$), and the constant speed phase ($T_2$). This assumption is consistent with the previous studies (*Domenici and Blake, 1991*; *Danos and*

*Lauder, 2012*; *Fleuren et al., 2018*) and was supported by our experiment (see *Figure 4—figure supplement 1*). Therefore:

$$T_{\mathbf{prey}} = T_1 + T_2 \tag{22}$$

For simplicity, the prey was assumed to end the fast-start phase at a certain displacement from the initial position in any $\alpha$ ($D_1$; the radius of the dotted circle in *Appendix 1—figure 1B*) and to move at a constant speed $U_{\mathrm{prey}}$ to cover the rest of the distance (toward the edge of the safety zone $\sqrt{X_{\mathrm{cross}}^2 + Y_{\mathrm{cross}}^2} - D_1$, plus the length of the body that is posterior to the CoM $L_{\mathrm{prey}}$). Because a larger $|\alpha|$ requires further turning prior to forward locomotion, which takes time (*Domenici and Blake, 1991*; *Ellerby and Altringham, 2001*), and the initial velocity after turning was dependent on $|\alpha|$ in our experiment (see *Figure 4B*), $T_1$ is given by a function of $|\alpha|$ ($T_1(|\alpha|)$). Therefore, $T_{\mathrm{prey}}$ can be expressed as:

$$T_{\mathbf{prey}} = T_1\left(|\boldsymbol{\alpha}|\right) + \frac{\sqrt{X_{\mathbf{cross}}^2 + Y_{\mathbf{cross}}^2} - D_1 + L_{\mathbf{prey}}}{U_{\mathbf{prey}}} \tag{23}$$

When the prey reaches the first part of the safety zone edge, $T_{\mathrm{pred}}$ can be expressed as:

$$T_{\mathbf{pred}} = \frac{D_{\mathbf{initial}} + D_2 + D_{\mathbf{react}} - X_{\mathbf{cross}}}{U_{\mathbf{pred}}} \tag{24}$$

When the prey reaches the second part of the safety zone edge, $T_{\mathrm{pred}}$ can be expressed as:

$$T_{\mathbf{pred}} = \frac{D_{\mathbf{react}} + R_{\mathbf{turn}}\tan^{-1}\frac{D_2\left(Y_{\mathbf{cross}} - R_{\mathbf{turn}}\right) - \left(X_{\mathbf{cross}} - D_{\mathbf{initial}} + D_{\mathbf{react}}\right)\left(D_{\mathbf{width}} - R_{\mathbf{turn}}\right)}{D_2\left(X_{\mathbf{cross}} - D_{\mathbf{initial}} + D_{\mathbf{react}}\right) + \left(D_{\mathbf{width}} - R_{\mathbf{turn}}\right)\left(Y_{\mathbf{cross}} - R_{\mathbf{turn}}\right)}}{U_{\mathbf{pred}}} \tag{25}$$

When the prey reaches the third or fourth part of the safety zone edge, $T_{\mathrm{pred}}$ can be expressed as:

$$T_{\mathbf{pred}} = \frac{D_{\mathbf{initial}} + D_{\mathbf{attack}}}{U_{\mathbf{pred}}} \tag{26}$$

From *Equations 1–16*, we can calculate $T_{\mathrm{diff}}$ in response to the changes of $\alpha$ and $\beta$, from $D_1$, $D_{\mathrm{width}}$, $D_{\mathrm{attack}}$, $R_{\mathrm{device}}$, $D_{\mathrm{initial}}$, $D_{\mathrm{react}}$, $R_{\mathrm{turn}}$, $U_{\mathrm{prey}}$, $U_{\mathrm{pred}}$, and $T_1(|\alpha|)$. Given that the escape success is assumed to be dependent on $T_{\mathrm{diff}}$, the theoretically optimal ET can be expressed as:

$$\mathbf{The\ optimal\ ET} = \underset{\boldsymbol{\alpha} + \boldsymbol{\beta}}{\mathbf{argmax}}\left(T_{\mathbf{diff}}\right) \tag{27}$$

