## [Editor Report]

This article will be of interest to researchers working on predator-prey interactions in the fields of biomechanics and neurosensory biology. It presents a valuable mathematical model that outputs possible escape trajectories given parameters relevant to the predator-prey system of interest. The premise of the modeling is attractive, as it includes the time required for prey to turn.

---

## [Decision Letter]

**Decision letter after peer review:**

Thank you for submitting your article "Geometric model incorporating prey's turn and predator attack endpoint explains multiple preferred escape trajectories" for consideration by *eLife*.

Your article has been reviewed by two peer reviewers, and the evaluation has been overseen by Christian Rutz as the Senior Editor. The following individual involved in the review of your submission has agreed to reveal their identity: Andrew D Bolton (Reviewer #1).

The reviewers have discussed their reviews with one another, and the Senior Editor has drafted this decision letter to help you prepare a revised submission.

Essential revisions:

Both reviewers were supportive of publication as long as a carefully revised submission clearly explains the rationale for how the speed of the predator was estimated and addresses the use of experimental parameters to model the escape response in other species. The measured values of predator speed should be reported. The additional points raised in the reviewers' full reports, which are appended below, also need to be addressed. Finally, please note that *eLife* has recently adopted the STRANGE framework, to help improve reporting standards and reproducibility in animal behaviour research – these recommendations are relevant to the empirical component of your study. In your revised submission, please consider the scope for sampling biases and potential limitations to the generalisability of your findings:

https://reviewer.elifesciences.org/author-guide/journal-policies

https://doi.org/10.1038/d41586-020-01751-5

*Reviewer #1 (Recommendations for the authors):*

Outside of my general public comments, a few thoughts. If you remove the predator adjustment model from Figure 1, Table 1 can be incorporated with the bottom left panel drawing. The bottom left of Figure 1 isn't as clear as it could be – the radius of the predator's mouth is certainly very subtle. Getting rid of the top right and bottom right panels (put them with the model's description in the Appendix) will free up explanatory space for the variables in the bottom left panel. The drawing could also use color to make each variable clear.

Figure 3 would benefit from 95% confidence intervals on the regression fit using bootstrap. Figure 4 needs a bit of work: 5 circles with a 3 ms aren't the right interval choice if the Tdiffs all max out at 10 -- it leaves the outer two circles with basically no information, and the max value is not on a div line. Figure 4A requires explanatory labeling for the concentric circles and axes; in fact, all circular plots in Figures 2 and 4 should have labels.

*Reviewer #2 (Recommendations for the authors):*

The predator's speed used to predict the optimal ET should be calculated from the empirical data for each interaction. Selecting a single predator speed for all interactions does not reflect the fact that the prey fish were responding to a stimulus that was approaching them at a particular speed.

Figure 1 is difficult to comprehend in its current format. The features of Figure 1C are the most important for understanding the model, but the formatting could be improved. For example, different colors could be used to define the different parameters and their labels. There also appears to be a mix of bitmap and vector elements here, the bitmap elements are low quality and upon magnification this becomes apparent. The caption refers readers to a table and an appendix in order to understand the different elements, which places a large burden on the reader. The figure and its caption should include all the information necessary for understanding.

It is unclear why the number of recordings of an individual prey fish was not standardized. This should be addressed along with more details of the experimental protocol that addresses the time between trials of an individual prey fish.

What was the procedure for selecting the particular kernel bandwidth value used to produce the results in Figure 2?

The data in Figure 2A are escape trajectories, what is the range of values that this variable can take? Is it 0-360, or -180 to 180? As described in the Methods section, it is unclear.

On away/toward responses: If Β = 180, then what is considered an away (toward) response? That is, what would the value of α be if the turn is CCW versus CW?

Figure 4: Since time difference is not a circular variable, rose plots are not appropriate for displaying these results.

[Editors’ note: further revisions were suggested prior to acceptance, as described below.]

Thank you for resubmitting your revised article entitled "Multiple preferred escape trajectories are explained by a geometric model incorporating prey's turn and predator attack endpoint" for further consideration by *eLife*. Your article has been evaluated by two reviewers and a Senior Editor.

The manuscript has been improved but there are some remaining issues that need to be addressed:

*Reviewer #1 (Recommendations for the authors):*

The authors have addressed nearly all of my comments. Figure 2 is much improved and no longer requires Table 1 to be in close proximity in order to understand the variables in the model. All requested fixes to plots have been accomplished and I remain in strong support of publication in *eLife*. However, I do believe that the authors misconstrued my argument about the stochastic nature of behavior in their paper, and how it fits into the bigger picture of stochastic behavior in animals generally. This may be due to my use of technical words during my review. To be clear, stochastic does not mean "completely random". A priori, this is a poor strategy to use when the location of a predator is accurately detected because it would inevitably result in unnecessary death a large percentage of the time. I also did not mean that the behavior should be noisy around the optimal choice; both of these strategies are presented in the new Figure 1. When I used the word "Bernoulli draw" in my review, what I meant is this. Drawing from a Bernoulli distribution is a stochastic binary choice. The prey fish is doing this during the behavior. They are drawing a "toward" versus "away" choice that the authors convincingly argue is parameterized by the probability mass dictated by their model-generated TDiff. In other words, if the "advantage" shown from the optimal to suboptimal peak is high, the probability of drawing an "away" swim becomes higher, while if the advantage is low, the choice of swimming toward or away becomes closer to equally likely. This is stochastic behavior, and it is probability matched to the topology of TDiff, which is an amazing finding, and likely leads to balanced unpredictability when playing the zero-sum game of escaping a predator who can predict trajectories if they are too stereotyped. If one were to directly translate the TDiff space into a continuous multimodal distribution, the fish could be thought of as drawing from this distribution stochastically and making the optimal choice when accounting for predator learning. This is what I meant in my review, and therefore the changes to Figure 1 don't really fit what I was trying to convey. My guess is that the Mauthner neurons of the fish are biased according to the Bernoulli probabilities dictated by the authors' model. It is fine with me if the authors want to leave Figure 1 as is to generally convey that the fish can use random strategies to circumvent predictability, and it is also fine if they choose to publish as is without my suggested reasoning. I just think that the idea of a Matching Law in ethological fish behavior is really interesting considering that most Matching Law results are in the context of unnatural reinforcement learning paradigms.

*Reviewer #2 (Recommendations for the authors):*

The authors addressed all concerns and suggestions in the revised manuscript. In particular, the new figure 1 and revised figure 2 convey the escape behavior and the geometric model much more clearly. The additional text describing predator speed optimization and the subsequent discussion add valuable context. Finally, the authors' revised text regarding their model's potential application to other species is more clear and better justified.

But I find that the authors are misinterpreting the findings of reference 42, Nair et al. In that paper, the stochastic strategy observed in zebrafish for initial orientations <30 (deg) and >150 (deg) was attributed to the distance advantage between an away and toward response. That is, when the advantage was small, either direction was equally likely. This is analogous to the authors' finding regarding the TDiff advantage. I think the authors should highlight the similarities in these findings and discuss how they relate to sensing the approaching predator.

---

## [Author Response]

Essential revisions:Both reviewers were supportive of publication as long as a carefully revised submission clearly explains the rationale for how the speed of the predator was estimated and addresses the use of experimental parameters to model the escape response in other species. The measured values of predator speed should be reported. The additional points raised in the reviewers' full reports, which are appended below, also need to be addressed.

As for the estimation of predator speed, (1) we added the explanation of why we used a fixed predator speed, and (2) we added the follow-up analysis of using the dummy predator speed per trial in the experiment. Please refer to the response to comment #2 from reviewer #2 for details.

The measured values of dummy predator speed per trial are added to Figure 5—figure supplement 2 (raw data is available in Dataset 1). The results show that the estimated value is slightly higher than the mean of the dummy predator speed (and the real predator speed as shown in the original version), suggesting that the value independently estimated in the present study is reasonable. The information is added in L286-288.

As for modeling the escape response in other species, we agree with reviewer #2 that the previous version was an over-interpretation. Therefore, we tempered our claims to clearly state that our model has only the potential to explain the different patterns of escape trajectories observed in previous works. Please refer to the response to comment #3 from reviewer #2 for details.

Finally, please note that eLife has recently adopted the STRANGE framework, to help improve reporting standards and reproducibility in animal behaviour research – these recommendations are relevant to the empirical component of your study. In your revised submission, please consider the scope for sampling biases and potential limitations to the generalisability of your findings:https://reviewer.elifesciences.org/author-guide/journal-policies

https://doi.org/10.1038/d41586-020-01751-5

Following the STRANGE framework, we added the following paragraphs and sentences in the Discussion and Materials and Methods.

The added paragraph in the Discussion (L530-537): Based on the STRANGE framework for animal behavior research [56], we identified potential biases that may limit the generalizability of our findings. Our empirical data are obtained from one species of hatchery-reared fish with a specific life stage, which has never experienced predators. Therefore, this study alone cannot exclude the possibility that fish of different species, origins, life stages, and rearing histories have different rules for ETs, which our model cannot explain. However, similar multiple preferred ETs have been observed in many fish species and other animal taxa, including hatcheries/wild origins and different life stages [26]. Therefore, we believe that our model is not specific to our experiment but is applicable to other cases showing multiple preferred ETs.

The added paragraph and sentences in Materials and Methods (L585-623): Following the STRANGE framework for animal behavior research [56], we provide details of the test samples and experimental procedure in the following two subsections.

*Sample Fish:* We used young-of-year juvenile hatchery-reared red sea bream *P. major* for the experiment. Sixty-five individuals were purchased from commercial hatcheries (Marua Suisan Co., Ltd., Ehime, Japan), where they were reared communally in artificial tanks. After arriving at the laboratory at Nagasaki University, they were kept in a 200 l polycarbonate tank and were fed with commercial pellets (Otohime C2; Marubeni Nisshin Feed Co. Ltd, Tokyo, Japan) twice a day. The sex of the fish was not determined because the species of this size is in a bisexual juvenile stage [61]. Water temperature was maintained at 23.8 to 24.9℃.

*Experimental Procedure:* We have elicited the escape response of *P. major* [45.3±3.5 (39.4–51.5) mm total length, 37.2±2.9 (32.3–42.2) mm standard length, 1.6±0.4 (0.9–2.3) g body weight, mean±s.d. (range), n=23] using a dummy predator. The value of Fulton’s condition factor [30.64±2.43 (26.10–35.56), mean±s.d. (range)], calculated by the body weight of the fish divided by the standard length cubed, suggests that all fish were in a good nutritious condition [62, 63]. The experiment was conducted in a plastic tank (540×890×200 mm) filled with seawater to a depth of 80 mm. The water temperature was maintained at 23.8 to 24.7℃. An individual *P. major* was randomly captured by a hand net from the holding tank, introduced into a PVC pipe (60 mm diameter) set in the center of the experimental tank, and acclimated for 15 min. Because it was not difficult to capture any individual by a hand net, there should be no bias in selecting individuals with specific behavioral types….(snip)… Each individual *P. major* was stimulated from 5 to 23 times with a time interval of at least 15 min, and, in total, 297 trials were conducted. We eliminated 33 trials from the analysis because *P. major* moved away from the striking course of the dummy predator before the stimulation (in 14 trials) and because bubbles obscured the *P. major* image (in 19 trials). The final data analyzed are 5–20 escape responses per individual and, in total, 264 escape responses. The experiments for each *P. major* were accomplished within one day to eliminate possible effects of tank transfer, handling, and change of rearing conditions. The number of recordings of an individual *P. major* was different because we could not allocate the same amount of time to the experiment per day due to the experimental schedule and because the numbers of eliminated data are different among individuals.

Reviewer #1 (Recommendations for the authors):Outside of my general public comments, a few thoughts. If you remove the predator adjustment model from Figure 1, Table 1 can be incorporated with the bottom left panel drawing. The bottom left of Figure 1 isn't as clear as it could be – the radius of the predator's mouth is certainly very subtle. Getting rid of the top right and bottom right panels (put them with the model's description in the Appendix) will free up explanatory space for the variables in the bottom left panel. The drawing could also use color to make each variable clear.

As mentioned in the response to comment #2, we incorporated the model variables into the panel (Figure 2B in the new version), made the drawing colored, and moved two panels (Figure 1B and D in the original version) to Appendix. Because Table 1 also includes the value and method for determining each variable, we would prefer to keep the table as is.

Figure 3 would benefit from 95% confidence intervals on the regression fit using bootstrap. Figure 4 needs a bit of work: 5 circles with a 3 ms aren't the right interval choice if the Tdiffs all max out at 10 -- it leaves the outer two circles with basically no information, and the max value is not on a div line. Figure 4A requires explanatory labeling for the concentric circles and axes; in fact, all circular plots in Figures 2 and 4 should have labels.

We added 95% Bayesian credible intervals to the figure (Figure 4 in the new version). Please note that we added Bayesian credible intervals instead of confidence intervals because this analysis was conducted by a hierarchical Bayesian method.

Regarding Figure 4 (Figure 5 in the new version), we changed the interval to 2.5 ms and made the maximum value on a division line.

We added labels for the concentric circles and axes for all circular plots of the figures (Figures 3 and 5 in the new version).

Reviewer #2 (Recommendations for the authors):The predator's speed used to predict the optimal ET should be calculated from the empirical data for each interaction. Selecting a single predator speed for all interactions does not reflect the fact that the prey fish were responding to a stimulus that was approaching them at a particular speed.

Please refer to the response to comment #2.

Figure 1 is difficult to comprehend in its current format. The features of Figure 1C are the most important for understanding the model, but the formatting could be improved. For example, different colors could be used to define the different parameters and their labels. There also appears to be a mix of bitmap and vector elements here, the bitmap elements are low quality and upon magnification this becomes apparent. The caption refers readers to a table and an appendix in order to understand the different elements, which places a large burden on the reader. The figure and its caption should include all the information necessary for understanding.

As mentioned in the response to comment #2 from reviewer #1, we added the model variables to the figure and made them colored (Figure 2B in the new version). We changed the low-quality bitmap elements to high-quality vector elements.

It is unclear why the number of recordings of an individual prey fish was not standardized. This should be addressed along with more details of the experimental protocol that addresses the time between trials of an individual prey fish.

The number of recordings of an individual prey fish was different for the following two reasons.

1. To eliminate possible effects of tank transfer, handling, and change of rearing conditions, we finished the experiment for each individual within a day. However, as we could not allocate the same amount of time to the experiment per day due to the experimental schedule, we could not conduct the same number of trials per individual.

2. We eliminated some of the data obtained from the analysis because the prey fish moved away from the striking course of the dummy predator before the stimulation in some cases and because bubbles obscured the prey fish image in other cases. These numbers of eliminated data are different among individuals.

The time between trials of an individual prey fish was at least 15 min. The above information and more details of the experimental protocol were added in L615-623.

What was the procedure for selecting the particular kernel bandwidth value used to produce the results in Figure 2?

We used a fixed kernel bandwidth value (50), which was manually selected after trying a variety of bandwidth values. We also tried automatic bandwidth selectors, but the results were unstable, suffering from overfitting and underfitting depending on different initial orientation bins and between the experiment and simulation. We would prefer to keep the kernel density curves because the curves facilitate the readers to view the multiple peaks in escape trajectories.

The data in Figure 2A are escape trajectories, what is the range of values that this variable can take? Is it 0-360, or -180 to 180? As described in the Methods section, it is unclear.

Because it is defined as the angular sum of *α* and *β* and because the range of |*β*| is 0~180° and the range of |*α*| was 9~147° in the experiment, the ET value can potentially range from −147° to 327°. To clarify this point, we added the following sentences and changed the axis tick value on the bottom from -90 to 270/-90 in all circular plots (Figures 3, 5, 6, and 7).

The added sentences (L579-582): ET is a circular variable with a cycle of 360°. As the range of |*β*| is 0~180° and the range of |*α*| was 9~147° in the experiment, the ET value can potentially range from −147° to 327°. Circular graphs are shown with angles from 0 to 360° [60], where negative values such as −90° correspond to positive values shifted by one cycle (in this case, −90°+360°=270°).

On away/toward responses: If Β = 180, then what is considered an away (toward) response? That is, what would the value of α be if the turn is CCW versus CW?

We thank the reviewer for pointing this out. This is a point that we did not mention in the previous version. These cases (β = 0 or 180°) were rare (1.1% in total), with two cases of β = 0° and one case of β = 180°. In such cases, the counterclockwise and clockwise turns were regarded as away and toward responses, respectively. We added the information in L570-572. Please also note that we changed the order for the explanation of variables accordingly (L563-572).

Figure 4: Since time difference is not a circular variable, rose plots are not appropriate for displaying these results.

We did not use rose plots but used circular plots to show the relationship between the escape trajectory (circular variable) and the relative time difference (non-circular variable). This kind of linear-circular relationship is often displayed by the circular plot with concentric circles as the axis of the linear variable, as we have done (e.g., Figure 5A in Martin et al. 2021). Therefore, we believe that the way we display these results is appropriate. To avoid misleading, we changed the sentence of the figure caption (Figure 5 in the new version) as follows: Relationship between the escape trajectory (ET) and the time difference between the prey and predator *T*_diff_ in different initial orientations *β*.

Cited reference: Martin BT, Gil MA, Fahimipour AK, Hein AM. Informational constraints on predator-prey interactions. Oikos. 2022;2022(10):e08143. doi: 10.1111/oik.08143.

[Editors’ note: further revisions were suggested prior to acceptance, as described below.]

Reviewer #1 (Recommendations for the authors):The authors have addressed nearly all of my comments. Figure 2 is much improved and no longer requires Table 1 to be in close proximity in order to understand the variables in the model. All requested fixes to plots have been accomplished and I remain in strong support of publication in eLife. However, I do believe that the authors misconstrued my argument about the stochastic nature of behavior in their paper, and how it fits into the bigger picture of stochastic behavior in animals generally. This may be due to my use of technical words during my review. To be clear, stochastic does not mean "completely random". A priori, this is a poor strategy to use when the location of a predator is accurately detected because it would inevitably result in unnecessary death a large percentage of the time. I also did not mean that the behavior should be noisy around the optimal choice; both of these strategies are presented in the new Figure 1. When I used the word "Bernoulli draw" in my review, what I meant is this. Drawing from a Bernoulli distribution is a stochastic binary choice. The prey fish is doing this during the behavior. They are drawing a "toward" versus "away" choice that the authors convincingly argue is parameterized by the probability mass dictated by their model-generated TDiff. In other words, if the "advantage" shown from the optimal to suboptimal peak is high, the probability of drawing an "away" swim becomes higher, while if the advantage is low, the choice of swimming toward or away becomes closer to equally likely. This is stochastic behavior, and it is probability matched to the topology of TDiff, which is an amazing finding, and likely leads to balanced unpredictability when playing the zero-sum game of escaping a predator who can predict trajectories if they are too stereotyped. If one were to directly translate the TDiff space into a continuous multimodal distribution, the fish could be thought of as drawing from this distribution stochastically and making the optimal choice when accounting for predator learning. This is what I meant in my review, and therefore the changes to Figure 1 don't really fit what I was trying to convey. My guess is that the Mauthner neurons of the fish are biased according to the Bernoulli probabilities dictated by the authors' model. It is fine with me if the authors want to leave Figure 1 as is to generally convey that the fish can use random strategies to circumvent predictability, and it is also fine if they choose to publish as is without my suggested reasoning. I just think that the idea of a Matching Law in ethological fish behavior is really interesting considering that most Matching Law results are in the context of unnatural reinforcement learning paradigms.

We highly appreciate the detailed explanation by the reviewer and apologize for misinterpreting the reviewer’s comment in the first round. We agree that the matching law, which has been formulated and tested in the field of psychology, is relevant to be mentioned in the paper. Because we believe that the new Figure 1 and the paragraph revised in the previous round of review (L49-66) will help understand the general patterns of escape trajectories (the reviewer 2 also found that it helped), we would like to keep them as is. To incorporate the idea of matching law, we added the following paragraphs, sentences, and phrases in the Introduction and Discussion.

The added paragraph in the Introduction (L73-84):

“Multiple preferred ETs of prey can result from situations in which animals choose one behavior from multiple options. Previous work carried out in the field of human and animal psychology on the choice of a particular behavioral strategy out of a number of options, has proposed a principle called “matching law.” According to this principle, the probability of a certain behavior to occur is related to the proportion of rewards obtained [30-33]. This is in contrast to a purely optimal tactic, where animals should always choose the best option (i.e., the highest rewards obtained) [33, 34]. Arguably, the field of predator-prey interactions has the potential to benefit from an analytical interpretation based on the matching law, because the multiple ETs available to the prey set a scenario similar to the multiple behavioral options considered in previous work analyzed using this principle. In line with this approach, the probability with which a prey chooses a particular escape trajectory can be related to the rewards (chances of survival) of each ET option calculated from a predator-prey geometric model.”

The added sentence/phrase in the Introduction (L93-100):

“First, using a fish species as a model, we tested whether our model could predict empirically observed multimodal ETs. Second, by calculating the chances of survival of each ET option from our model, we investigated how the prey fish chose a given ET from multiple options. Third, by extending the model, we tested whether other patterns of empirical ETs could be predicted: unimodal ETs and multimodal ETs directed at small (20–50°) and large (150–180°) angles from the predator's approach direction. The biological implications resulting from the model and experimental data are then discussed within the frameworks of predator-prey interactions and behavioral decision-making.”

The added paragraph in the Discussion (L513-523):

“From a behavioral decision-making perspective, our results suggest that the prey follows the matching law [30-33], where the probability that an optimal or suboptimal ET is chosen is proportional to its chances of survival (i.e., *T*_diff_). As the matching law predicts [33], the prey stochastically draws from a Bernoulli distribution dictated by the optimal ET advantage for the binary choice between an optimal or suboptimal ET, thereby introducing an element of unpredictability, which can prevent predators from learning. Because most empirical studies supporting the matching law use unnatural reinforcement learning paradigms or human behaviors [30-33], this result suggests that the matching law is also applicable to animal behavior in realistic contexts. Further research using a real predator and dummy prey (e.g., [53]) controlled to escape toward an optimal or suboptimal ET with various specific probabilities is required to test whether our model accurately predicts the best combination of the optimal and suboptimal ETs when accounting for the predator learning.”

The added sentence in the Discussion (L530-532):

“More specifically, the Mauthner cell and other neurons involved may be activated in accordance with the Bernoulli probabilities dictated by the model, which determine the proportions of away and toward responses and the magnitude of turn to achieve the multiple preferred ETs.”

Reviewer #2 (Recommendations for the authors):The authors addressed all concerns and suggestions in the revised manuscript. In particular, the new figure 1 and revised figure 2 convey the escape behavior and the geometric model much more clearly. The additional text describing predator speed optimization and the subsequent discussion add valuable context. Finally, the authors' revised text regarding their model's potential application to other species is more clear and better justified.But I find that the authors are misinterpreting the findings of reference 42, Nair et al. In that paper, the stochastic strategy observed in zebrafish for initial orientations <30 (deg) and >150 (deg) was attributed to the distance advantage between an away and toward response. That is, when the advantage was small, either direction was equally likely. This is analogous to the authors' finding regarding the TDiff advantage. I think the authors should highlight the similarities in these findings and discuss how they relate to sensing the approaching predator.

We appreciate the reviewer for pointing out this issue. We revised the parts first to mention the similarities in the findings between our results and those of Nair et al., and then how they relate to sensing the approaching predator.

The revised sentences (L499-503) are as follows:

“The unpredictability at initial orientations near 0° and 180° is consistent with the study that applied the conventional geometric model to the larval zebrafish *Danio rerio* [47], where the optimal and suboptimal ETs are approximately symmetrical to the axis of the predator attack. This phenomenon can be explained by the toward-away indecision at orientations nearly perpendicular to the threat [28, 52].”